# Apoptotic Effect of Gallic Acid via Regulation of p-p38 and ER Stress in PANC-1 and MIA PaCa-2 Cells Pancreatic Cancer Cells

**DOI:** 10.3390/ijms242015236

**Published:** 2023-10-16

**Authors:** Jeong Woo Kim, Jinwon Choi, Moon Nyeo Park, Bonglee Kim

**Affiliations:** 1Department of Pathology, College of Korean Medicine, Kyung Hee University, Hoegidong Dongdaemungu, Seoul 05253, Republic of Korea; jeongwoo21@khu.ac.kr (J.W.K.); wlsdnjsl6888@naver.com (J.C.); mnpark@khu.ac.kr (M.N.P.); 2Korean Medicine-Based Drug Repositioning Cancer Research Center, College of Korean Medicine, Kyung Hee University, Hoegidong Dongdaemungu, Seoul 05253, Republic of Korea

**Keywords:** pancreatic cancer, gallic acid, apoptosis, ER stress, p38 pathway, ROS

## Abstract

Pancreatic cancer (PC) is currently recognized as the seventh most prevalent cause of cancer-related mortality among individuals of both genders. It is projected that a significant number of individuals will succumb to this disease in the forthcoming years. Extensive research and validation have been conducted on both gemcitabine and 5-fluorouracil as viable therapeutic options for PC. Nevertheless, despite concerted attempts to enhance treatment outcomes, PC continues to pose significant challenges in terms of achieving effective treatment alone through chemotherapy. Gallic acid, an endogenous chemical present in various botanical preparations, has attracted considerable attention due to its potential as an anticancer agent. The results of the study demonstrated that gallic acid exerted a decline in cell viability that was dependent on its concentration. Furthermore, it efficiently suppressed cell proliferation in PC cells. This study observed a positive correlation between gallic acid and the production of reactive oxygen species (ROS). Additionally, it confirmed the upregulation of proteins associated with the protein kinase-like endoplasmic reticulum kinase (PERK) pathway, which is one of the pathways involved in endoplasmic reticulum (ER) stress. Moreover, the administration of gallic acid resulted in verified alterations in the transmission of mitogen-activated protein kinase (MAPK) signals. Notably, an elevation in the levels of p-p38, which represents the phosphorylated state of p38 MAPK was detected. The scavenger of reactive oxygen species (ROS), N-Acetyl-L-cysteine (NAC), has shown inhibitory effects on phosphorylated p38 (p-p38), whereas the p38 inhibitor SB203580 inhibited C/EBP homologous protein (CHOP). In both instances, the levels of PARP have been successfully reinstated. In other words, the study discovered a correlation between endoplasmic reticulum stress and the p38 signaling pathway. Consequently, gallic acid induces the activation of both the p38 pathway and the ER stress pathway through the generation of ROS, ultimately resulting in apoptosis. The outcomes of this study provide compelling evidence to support the notion that gallic acid possesses considerable promise as a viable therapeutic intervention for pancreatic cancer.

## 1. Introduction

Cancer continues to be a formidable disease that has not yet been effectively eradicated. PC is associated with a high mortality rate, resulting in nearly equal numbers of deaths (466,000) and cases (496,000). This poor prognosis positions pancreatic cancer as the seventh most common cause of cancer-related mortality in both males and females [1]. According to projections, it is anticipated that by the year 2025, a total of 111,500 individuals will succumb to the effects of a certain condition, which shall be referred to as PC [2]. The negative prognosis of this disease can be ascribed to its prompt systemic dissemination and vigorous local proliferation [3]. In the case of patients diagnosed with resectable PC, the customary strategy entails surgical intervention as the first therapeutic measure, subsequently complemented by adjuvant chemotherapy [4]. The efficacy of gemcitabine and 5-fluorouracil in the treatment of pancreatic cancer has been confirmed, with gemcitabine being the preferred choice due to its better tolerability in the setting of chemotherapy following the surgical removal of the tumor [5]. In contemporary medical practice there has been a notable trend towards the concurrent administration of nab-paclitaxel and gemcitabine [6]. The efficacy of gemcitabine and other therapeutic medications in the treatment of advanced and metastatic PC has been demonstrated. However, the emergence of chemoresistance to gemcitabine poses a substantial obstacle to the success of this chemotherapy. PC cells demonstrate a greater level of resistance to gemcitabine in comparison to other chemotherapeutic agents [7]. The current chemotherapeutic options demonstrate restricted effectiveness in the treatment of pancreatic cancer.

Two cell lines, namely undifferentiated carcinoma MIA PaCa-2 and poorly differentiated adenocarcinoma PANC-1, were utilized to illustrate the early morphological changes in the peritoneal metastasis (PM). Furthermore, the connection between these morphological dynamics and their oncogenic properties was examined [8,9]. Hence, the primary objective of this work was to examine the apoptotic effects induced by gallic acid and clarify the underlying mechanisms via which it exerts its anticancer properties.

Herbal medicines have been increasingly prevalent as a potentially effective treatment for chronic illnesses, such as cancer, in diverse global contexts, particularly in China, Japan, Egypt, and India [10]. The chemical known as gallic acid, which is naturally present in a range of herbal preparations, has attracted considerable attention due to its potential as an anticancer agent. Gallic acid is present in a variety of botanical sources, such as *Toona sinensis*, *Punica granatum*, *Quercus frainetto*, *Terminalia bellirica*, and grape seeds [11,12,13,14,15]. The anticancer effects of gallic acid were observed in oral squamous carcinoma, as evidenced by the down-regulation of survivin and clAP1 [11]. Moreover, it was observed that gallic acid demonstrated inhibitory effects on the synthesis of nitric oxide (NO), prostaglandin E-2 (PGE-2), and interleukin-6 (IL-6) in RAW264.7 cells when stimulated with lipopolysaccharide (LPS) [12]. Gallic acid has been demonstrated to exhibit a range of advantageous characteristics, such as anti-inflammatory, antiproliferative, antitumorigenic, and adipocyte differentiation inhibitory effect [12,14,15].

The ER is a cellular organelle that plays a crucial role in protein folding and signaling processes. It is worth noting that excessive ER stress has the potential to trigger apoptosis, a programmed cell death mechanism [16]. The activation of p38 MAPK (Mitogen-Activated Protein Kinase) facilitates apoptosis by facilitating the relocation of the proapoptotic protein Bax from the cytoplasmic compartment to the mitochondrial compartment [17]. The ER stress pathway and the p38 MAPK pathway are frequently modulated by ROS. Recent research indicates that ROS-mediated ER stress through the p38 activation pathways can induce apoptosis in cervical cancer [18]. The findings indicate a significant correlation between the p38 pathway and ER stress.

Gallic acid has been employed in a diverse range of medicinal treatments [19,20,21,22,23,24,25]. Nevertheless, there is a lack of research undertaken on the relationship between ER stress and p38-mediated apoptosis, specifically in the pancreatic cancer cell lines, PANC-1 and MIA PaCa-2. In this study, the apoptotic effects induced by gallic acid clarify the underlying mechanisms via which it exerts its anticancer properties in PANC-1 and MIA PaCa-2 cells.

## 2. Results

### 2.1. Gallic Acid Cytotoxic and Antiproliferative in Cancer Cells PANC-1 and MIA PaCa-2 While Not Affecting Normal Cell Line, L-929

In order to assess the cytotoxic and antiproliferative effects of gallic acid, cell viability assay, and colony formation assay were employed in multiple cancer cell lines. In this study, it was shown that gallic acid exhibited inhibitory effects on the PANC-1 and MIAPaCa-2 cell lines at varying concentrations. However, no significant effects of gallic acid were observed on the normal cell line such as L-929, as depicted in Figure 1B–D. The maximum concentration was set to 300 µM and diluted continuously (18.75, 37.5, 75, 150, 200, 250 and 300 µM) (Figure 1B–D). Based on the cell viability assay results, the experimental concentration was set to 75 µM and 150 µM. In a colony formation assay, it was shown that the proliferation of PANC-1 and MIAPaCa-2 cells was suppressed by gallic acid (Figure 1E,H).

### 2.2. Gallic Acid Increased the Sub-G1 Ratio and Induced Apoptosis in Pancreatic Cancer Cells

In order to determine the underlying mechanisms behind the cytotoxic and antiproliferative effects of gallic acid, this study employed TUNEL and cell cycle analysis procedures on the PANC-1 and MIA PaCa-2 cell lines (Figure 2). The nucleus is visualized through blue fluorescence, while green fluorescence indicates apoptosis resulting from DNA fragmentation. The observed trend indicated a negative correlation between concentration and cell count, as well as a positive correlation between concentration and DNA fragmentation (Figure 2A). Here, GA increased the sub-G1 population in pancreatic cancer cells PANC-1 and MIA PaCa-2 (Figure 2D,E).

### 2.3. Gallic Acid Regulated the Expression of Apoptosis-Related Proteins

Western blot analysis was conducted in order to validate the presence of proteins associated with apoptosis (Figure 3A,B). Consequently, the expression of Pro-PARP was reduced, while the levels of cleaved PARP, caspase-3, Bax, and p-histone H2A.X were elevated in a concentration-dependent manner (Figure 3).

### 2.4. Gallic Acid Caused ROS Production and ER Stress-Mediated Apoptosis in PANC-1 and MIA PaCa-2 Cells

The identification of ROS generation and ER stress-related apoptosis was conducted using an ROS detection kit and Western blotting analysis. The verification of ROS production was conducted by the utilization of 2′,7′-dichlorofluorescin diacetate (DCFDA) staining. The gallic acid resulted in an increase in ROS production in pancreatic cancer cell lines, including PANC-1 and MIA PaCa-2 cells (Figure 4A,B). However, this treatment did not have any discernible effect on normal cells, such as L-929 cells (Figure 4C). The induction of PERK, ATF4, and CHOP by the gallic acid in a concentration-dependent manner was seen in the PANC-1 and MIA PaCa-2 cells (Figure 4D,E). In a concentration-dependent manner, the levels of nitric oxide (NO) continuously exhibited an upward trend upon exposure to gallic acid (Figure 4H,I).

### 2.5. Gallic Acid Induced p38 Pathways-Mediated Apoptosis in PANC-1 and MIA PaCa-2 Cells

The study employed western blotting as a method to examine the MAPK signaling pathways that were triggered by gallic acid in pancreatic cancer cells. The results revealed that gallic acid induced apoptosis by activating the expression of p38. However, there was no observed modification in the activation of ERK and JNK expression in PANC-1 and MIA PaCa-2 cells (Figure 5A–D).

### 2.6. ROS/p-p38 Signal Pathway Has Pivotal Roles in Gallic Acid Induced Apoptosis

The confirmation of the association between the p38 pathway and ER stress was achieved through the pretreatment of NAC, a ROS scavenger, SB203580, and a p38 inhibitor, in the PANC-1 and MIA PaCa-2 cell lines. The treatment of the 5 mM concentration of NAC for 2 h led to a decrease in ROS generation caused by the presence of 150 μM gallic acid (Figure 6A,B). The western blot assay was employed using identical conditions for experimentation, revealing a decrease in the levels of phosphorylated p38 (p-p38) following treatment with NAC (Figure 6C,D). After a pretreatment period of 1 h with a concentration of 20 μM SB203580, a decrease in the increased levels of CHOP was detected (Figure 6E,F). Furthermore, the levels of poly (ADP-ribose) polymerase (PARP), which had experienced a decrease in both instances, underwent restoration.

## 3. Discussion

Pancreatic cancer represents significant challenges in terms of curability and exhibits a low survival rate. Pancreatic cancer ranks as the seventh leading cause of cancer-related mortality among both genders, resulting in a significant number of fatalities. In order to combat cancer, numerous investigations have been conducted. The compound known as kaempferol has been observed to effectively scavenge ROS within pancreatic cancer cells, hence providing evidence for its potential as an anticancer agent [26]. The polyphenolic chemical agrimoniin has been demonstrated to exhibit a potent cytotoxic effect on several cancer cell types through the induction of energy metabolism malfunction mediated by ROS [27]. Gallic acid, an organic chemical of phenolic nature obtained from botanical sources, exhibits a wide range of therapeutic characteristics. Several studies have provided evidence that gallic acid had anticancer properties by acting through various biological mechanisms. These mechanisms include impeding cell migration and metastasis, facilitating programmed cell death (apoptosis), causing cell cycle arrest, restraining the formation of new blood vessels (angiogenesis), and regulating the expression of oncogenes [19]. As previously stated, it has been demonstrated that gallic acid possesses the ability to inhibit diverse cellular mechanisms implicated in the onset, advancement, and advancement of cancer in numerous malignancies [28].

Thus, we evaluated the potential applicability of gallic acid in pancreatic cancer therapy. Specifically, we examined the apoptotic mechanism of gallic acid and its relationship with ROS-mediated p38 activation signaling. Here, it was shown that the treatment of gallic acid resulted in a notable decrease in cell viability in PANC-1 and MIA PaCa-2 cells compared to L-929 normal cells. This suggests that gallic acid exhibits a strong antitumor effect specifically targeting cancer cells. Similarly, the proliferation of PANC-1 and MIA PaCa-2 cells was inhibited by gallic acid, as demonstrated using the colony formation experiment (Figure 1).

Two flow cytometric techniques were employed to evaluate endonucleolysis, which occurred concurrently with apoptosis produced by these substances. One method quantified DNA content following the extraction of low molecular weight DNA, while the other method utilized exogenous terminal deoxynucleotidyl transferase to mark DNA strand breaks in situ. The identified approaches were able to detect apoptotic cells and examine the distribution of the cell cycle in the unaffected cell population. Additionally, the use of terminal transferase allowed for the identification of apoptotic cell cycle positions [29]. The propidium iodide (PI) flow cytometric method has been extensively employed to evaluate apoptosis in many experimental contexts since its conception [30]. Inflow cytometric analyses that only employ propidium iodide (PI) labeling, apoptotic cells typically exhibit hypodiploidy [31]. Notably, a biological investigation demonstrates that anticancer activity effectively suppresses the growth of various cancer cells by producing a subG1 population and promoting apoptosis [32,33,34,35,36,37,38,39]. The reduced DNA content observed in sub-G0/G1 cells can be attributed, in part, to the depletion of fragmented chromatin, which is widely recognized as a characteristic feature of apoptosis [40]. Consistently, the arrest of the Sub-G1 population is indicative of the occurrence of apoptosis [41].

As depicted in Figure 2, it is evident that gallic acid consistently induced an increase in TUNEL-positive cells and the sub-G1 arrest, indicating the apoptotic potential of gallic acid in PANC-1 and MIA PaCa-2 cells.

Both processes result in the initiation of a series of proteases known as caspases (cysteine-containing aspartic acid-specific proteases). These caspases are first produced as inactive proenzymes and are activated through proteolytic cleavage. This activation may also lead to the cleavage of other caspases, so contributing to the apoptotic signaling cascade [42,43,44,45,46]. Upon being activated by pro-apoptotic signals, initiator caspases (such as Caspase 2, 8, 9, and 10) cleave and subsequently activate effector caspases (such as Caspase 3, 6, and 7) [47,48,49,50,51,52,53]. Thus, the depolarization of mitochondria and subsequent activation of the caspase cascade, specifically caspase-3, were seen. Additionally, the cleavage of PARP exhibited a dose-dependent increase, which is indicative of apoptosis [54]. Notably, the nuclear enzyme poly (ADP-ribose) polymerase facilitates the conversion of NAD+ into poly(ADP-ribose). The protein being investigated possesses a DNA-binding domain at its N-terminus, consisting of two zinc fingers. The NAD+-binding domain located at the C-terminal region undergoes modification through the addition of a brief sequence containing many glutamic acid residues, resulting in auto-poly(ADP-ribosyl)ation. The inherent propensity of unmodified polymerase molecules towards DNA strand breaks is counteracted by the process of auto-poly(ADP-ribosyl)ation, which leads to the dissociation of the protein. This dissociation enables DNA repair enzymes to access and repair the damaged regions [55]. The serine 139 phosphorylation of histone H2AX is regarded as a first cellular reaction in human cells when exposed to DNA breaks caused by ionizing radiation. DNA breaks also occur during the final phases of apoptosis, wherein chromosomal DNA is fragmented into oligonucleosomal fragments [56]. The observed elevation in levels of phospho-Histone H2A.X and BAX indicates the occurrence of apoptosis. Survivin is commonly observed to be present in elevated quantities within cancerous cells, while a reduction in survivin expression signifies the onset of apoptosis [57]. This study suggests that gallic acid has the potential to induce apoptosis in PANC-1 and MIA PaCa-2 cells through a concentration-dependent mechanism. This is supported by the observed rise in cleaved PARP, caspase 3, Bax, and pH2AX, as well as the attenuation of pro PARP and survivin. These findings may indicate that gallic acid induces apoptosis via caspase-mediated pathways (Figure 3).

Under typical conditions, ROS produced by mitochondrial oxidative respiration stimulate a second messenger in the Ca^2+^-mediated cascade. Meanwhile, ROS-mediated cancer cell death is associated with ongoing ER stress due to increased ROS production in cancer cells. Apoptosis-mediated cell death can be induced by a number of naturally occurring chemicals via altering ROS production [58,59]. During severe stress in the ER, the activated PERK phosphorylates eIF2α, leading to the activation of ATF4. The transcription factor ATF4 induces the expression of the CHOP gene and worsens stress through ROS generation [60]. The ROS encompass a range of chemical entities, such as the superoxide anion (O_2_^−^), hydrogen peroxide (H_2_O_2_), hydroxyl radical (OH^−^), singlet oxygen (^1^O_2_), and ozone (O_3_) [54,61]. Furthermore, other additional reactive species participate in redox signaling, including nitric oxide and hydrogen sulfide [62]. Gallic acid is a widely recognized chemical, and its anti-tumor actions on various types of malignancies have been substantiated through the induction of ROS [63,64,65,66,67]. Low NO concentrations frequently induce proliferation, metastasis, EMT, angiogenesis, invasion, and migration of breast cancer cells. Elevated levels of NO inhibit the proliferation of cancer cells and facilitate apoptosis [68]. Furthermore, the interplay between NO and ROS can result in elevated levels of NO and peroxynitrite, which have the ability to directly induce apoptosis in tumor cells. The aforementioned fact has led to the development of various therapeutic strategies, involving the utilization of exogenous NO donors as chemotherapeutic agents [69,70,71,72,73,74,75,76,77]. Gallic acid-induced ROS-dependent NO generation in pancreatic cancer cells, suggests that it may be used as a therapeutic agent for inducing apoptosis in pancreatic cancer (Figure 4).

Mitogen-activated protein kinases (MAPKs) are a class of kinases that are integral in the regulation of diverse biological processes. The MAP kinase family is composed of four primary sub-families of interconnected proteins, including extracellular regulated kinase 5 (ERK5), extracellular regulated kinases 1/2 (ERK1/2), c-Jun N-terminal kinase (JNK), and p38 [78]. Moreover, the p38 mitogen-activated protein kinase (MAPK) pathway serves a crucial function in the transmission of stress signals originating from the external environment. It is recognized as possessing dual functionality, acting as both a suppressor and promoter of tumorigenesis [79,80]. The presence of KRAS mutation in around 95% of pancreatic ductal adenocarcinoma (PDAC) cases has led to investigations on the relationship between p-p38 overexpression in PDAC cancer cells and patient survival [81]. It has been observed that higher expression levels of p-p38 are associated with longer survival, in contrast to cases with lower expression levels. Hence, the protein p-p38 serves as a reliable prognostic indicator for PDAC [81]. Recently, a new approach has been established to produce agents with novel anticancer characteristics by activating p38 MAPK for the treatment of pancreatic cancer [82]. Interestingly, it has been hypothesized that the development and utilization of chemicals capable of altering the intracellular levels of NO and ROS could potentially be an effective strategy for disturbing the redox equilibrium that is active in cancer cells [83]. Therefore, the current study examined the apoptotic mechanism of gallic acid in conjunction with the NO or ROS-mediated p38 MAPK activation signaling to determine whether or not gallic acid could be used in the treatment of pancreatic cancer. The findings consistently demonstrate that the production of ROS and NO, as well as the activation of p38, were elevated in the PANC-1 and MIA PaCa-2 cells following treatment with gallic acid. These results suggest that gallic acid disrupts the redox equilibrium potential in pancreatic cancer cells (Figure 5).

Furthermore, the anticancer effect shown in several types of cancer cells was facilitated through CHOP-dependent apoptosis [54,59,84,85,86,87,88,89]. The CHOP protein, which is linked to endoplasmic reticulum (ER) stress, plays a pivotal role in initiating apoptosis in cells under sustained intracellular stress caused by the generation of ROS [54,61].To further authenticate, it was established that the administration of a reactive oxygen species (ROS) inhibitor (NAC) efficiently suppressed the generation of ROS. Furthermore, it was shown that the introduction of gallic acid resulted in an increased expression of CHOP, which was achieved by employing a phospho-p38 inhibitor (SB203580) and a ROS inhibitor (NAC). Therefore, the findings of the above studies provide confirmation that gallic acid exerts an influence on apoptosis through the involvement of ROS-induced CHOP, that and the p-p38 (Figure 6).

The main purpose of this study was to investigate the effect of gallic acid on PANC-1 and MIA PaCa-2 pancreatic cancer cell lines. Nevertheless, the current evidence is inadequate to establish a comprehensive explanation or mechanism. The results given in this study are derived from in vitro experiments conducted on pancreatic cancer cell lines. However, it is important to note that additional research and validation through in vivo studies and clinical trials are necessary to determine the relevance and application of these findings to the human body and therapeutic settings. Hence, it is imperative to do additional research in subsequent studies.

Consequently, it can be inferred that gallic acid induces activation of both the p38 pathway and the ER stress pathway through the generation of ROS, eventually resulting in apoptosis (Figure 7).

## 4. Materials and Methods

### 4.1. Chemicals and Reagents

The gallic acid (M.W. = 170.12; CAS No. 149-91-7) was purchased from Sigma-Aldrich (St. Louis, MO, USA). Gallic acid stock at 300 mM in ethanol was stored at −20 °C. Propidium iodide (PI), DCF-DA, 4′6-diamidino-2-phenylindole (DAPI), and N-acetyl-L-cysteine (NAC) were purchased from Sigma-Aldrich (St. Louis, MO, USA). p38 inhibitor (SB203580) was purchased from Selleck (Houston, TX, USA).

### 4.2. Cell Culture

Pancreatic cancer cell lines PANC-1 and MIA PaCa-2 and normal cell line (L-929) were purchased from the Korean Cell Bank (Seoul, Republic of Korea). PANC-1 and MIA PaCa-2 cells were cultured with DMEM high glucose containing 10% FBS and 1% penicillin/streptomycin (Biowest, Nuaillé, France). L-929 cells were grown using RPMI 1640 media supplemented with 10% fetal bovine serum (FBS), 2 μM L-glutamine, and 10,000 U/mL of Penicillin/streptomycin (Biowest, Nuaillé, France). Cells were incubated at 37 °C and humidified in 5% CO_2_ conditions (MCO-15AC, Sanyo, Osaka, Japan). Cell subculture was conducted at regular intervals of 2–3 days.

### 4.3. Cell Viability Assay

PANC-1, MIA PaCa-2 and L-929 cells were seeded at a density of 1.0 × 10^4^ cells per well in a 96-well plate and incubated for 24 h. The cell viability assay was conducted using DMEM and RPMI supplemented with 10% fetal bovine serum (FBS) and 1% penicillin/streptomycin. Cells were treated with various concentrations of gallic acid (18.75, 37.5, 75, 150, 200, 250 and 300 µM) for 24 h. Then, EZ-CYTOX cell viability assay reagent (Daeil Lab Service, Seoul, Republic of Korea) was used according to the manufacturer’s manual. The absorbance was measured at 450 nm by a microplate reader (Bio-Rad, Hercules, CA, USA).

### 4.4. Colony Formation Assay

PANC-1 and MIA PaCa-2 cells were seeded at a density of 5.0 × 10^3^ cells/well in a 6-well plate and incubated for 24 h. The cells were subjected to a 24 h treatment with gallic acid. Over the course of time, fresh media was added. PANC-1 cells were subjected to an incubation period of 9 days, while MIA PaCa-2 cells were incubated for a duration of 8 days. The cells underwent a washing step using DPBS, followed by fixation and staining using Diff-Quik solution (Sysmex, Kobe, Japan). The quantification of colonies was conducted using the ImageJ Version 1.53i.

### 4.5. TUNEL Assay

PANC-1 and MIA PaCa-2 cell lines were cultured on a 4-well culture slide (SPL, in Pocheon, Republic of Korea). After an overnight incubation, the gallic acid was applied onto the slide glass for 24 h and afterwards washed with phosphate-buffered saline (PBS). The cells underwent fixation by treatment with a 4% paraformaldehyde solution (Bylabs, San Francisco, CA, USA) and were subsequently rinsed with PBS. The specimens were subsequently permeabilized using a solution containing 0.2% Triton X-100 (Promega, Madison, WI, USA) and afterwards rinsed with phosphate-buffered saline (PBS). The DeadEndTM Fluorometric TUNEL System kit (Promega, Madison, WI, USA) was employed for cell staining. Following a 10 min treatment of the equilibrium buffer, the Nucleotide Mix and rTdT enzyme should be combined and incubated for a 1 h. The 2× SSC solution was subjected to a 15 min reaction time and afterwards rinsed with PBS. Subsequently, the specimen was subjected to staining at a concentration of 1 μg/mL of DAPI, followed by mounting using a solution specifically designed for this purpose. The sample was observed using a confocal microscope Zeiss LSM 800 (Zeiss, Oberkochen, Germany).

### 4.6. Cell Cycle Analysis

PANC-1 and MIA PaCa-2 cell lines were subjected to treatment with gallic acid at concentrations of 75 and 150 µM for 24 h. Cells were harvested with Trypsin-EDTA solution (Welgene, Gyeongsangbuk-do, Gyeongsan, Republic of Korea) and fixed with 70% cold ethanol for overnight at −20 °C. The cells were treated with RNase A (1 mg/mL) for 50 min at 37 °C. After that, the cells were stained with propidium iodide (50 µg/mL) for 2 h. Cell cycle analysis (FL-2) was performed with the FACSCalibur (Becton Dickinson, Bergen, Franklin Lakes, NJ, USA).

### 4.7. Western Blot Analysis

PANC-1 and MIA PaCa-2 cells were seeded at a density of 2.0 × 10^5^ cells per well in a 6-well plate and treated with 75 and 150 µM gallic acid for 24 h. Afterward, the cells were harvested using a Trypsin-EDTA solution, cells were lysed with proEXTM CETi lysis buffer with phosphatase inhibitors (Translab, Daejeon, Republic of Korea). It was disassembled using Vortex for 1 min and stabilized in ICE for 10 min. This process was repeated three times. After collecting the supernatant from cell lysate, the same amount of protein was calculated through a BCA assay. The protein of each sample was mixed with 5× loading dye and distilled water and then heated at 100 °C for 5 min. The proteins in each lane were loaded onto the SDS-PAGE gels with varying concentrations of 8–12% at 95 V for 110 min. These proteins were subsequently transferred to polyvinylidene fluoride membranes (PVDF, Millipore, Burlington, MA, USA). Following the transfer process, it is necessary to examine the protein using Ponceau S staining. The membranes underwent three rounds of washing, each lasting 5 min, using TBST solution containing 0.1% Tween 20. Subsequently, the membranes were blocked for a duration of 1 h using a 5% skimmed milk solution that was liquefied in TBST. The membranes underwent a washing process using TBST solution, followed by immersion in specific primary antibodies for an extended period of time; β-actin (1:1000), (Santa Cruz Biotechnology, Dallas, TX, USA), PARP (1:1000), cleaved caspase3 (1:1000), Phospho-Histone H2A.X (1:1000), Bax (1:1000), survivin (1:1000), PERK (1:1000), CHOP (1:1000), Phospho-p44/42 MAPK(Erk1/2) (1:1000), Phospho-SAPK/JNK(Thr183/Tyr185) (1:1000), Phospho-p38 MAPK (1:1000), p38 MAPK (1:1000) (Cell Signaling, Beverly, MA, USA), ATF4 (1:10,000) (Abcam, Cambridge, UK). After 30 min of TBST washing, the membranes were incubated at room temperature for 2 h with HRP-conjugated secondary anti-mouse or anti-rabbit IgG. Prior to protein analysis, the membranes underwent three washes with TBST. Membranes were visualized by chemiluminescence imaging equipment (Davinch-K, Seoul, Republic of Korea) by processing ECL select Western blotting detection reagent (Cytiva, Seoul, Republic of Korea).

### 4.8. ROS Assay

PANC-1, MIA PaCa-2 and L-929 cells were seeded at a density of 1.0 × 10^4^ cells per well in a 96-well plate and incubated for 24 h. DCFDA (20 µM) (Sigma Aldrich, St. Louis, MO, USA) was used to stain cells at 37 °C for 45 min. Each well washed and treated the gallic acid (75, 150 µM) for 4 h with DMEM high glucose containing 10% FBS and no phenol red (Sigma Aldrich, St. Louis, MO, USA). The fluorescence was measured at 450 nm, 535 nm by a microplate reader (Thermo Fisher, Vantaa, Finland).

### 4.9. Nitric Oxide Assay

PANC-1 and MIA PaCa-2 cells (2.0 × 10^5^ cells/well) were seeded and treated for 24 h with 75 and 150 µM gallic acid in 6-well plate. Using a Trypsin-EDTA solution, cells were harvested and then centrifuged. After that, the supernatant was separated by a 0.22 µm filter. The cell-free supernatant was transferred to the 96 well plate, and it was analyzed using a nitric oxide assay kit (iNtRON Partners, Gyeonggi-do, Seongnam, Republic of Korea).

### 4.10. N-Acetylcysteine (NAC) Assay

In a 96-well plate, PANC-1 and MIA PaCa-2 cells (1.0 × 10^4^ cells/well) were seeded for 24 h. After pretreatment with 5 mM NAC for 2 h, cells were stained for 45 min at 37 °C with DCFDA (20 µM) (Sigma Aldrich, St. Louis, MO, USA). Washing each well and treating gallic acid (75, 150 µM) for 4 h with DMEM high glucose containing 10% FBS and no phenol red (Sigma Aldrich, St. Louis, MO, USA).

### 4.11. N-Acetylcysteine (NAC) Study

PANC-1 and MIA PaCa-2 cells were seeded at a density of 2.0 × 10^5^ cells per well in a 6-well plate and incubated for 24 h. Following a pretreatment period of 2 h with a concentration of 5 mM NAC, gallic acid was subsequently treated with a concentration of 150 μM. Following that, the stability of pro-PARP and p38 was assessed by Western blotting.

### 4.12. p 38 Inhibitor Study

In a 6-well plate, PANC-1 and MIA PaCa-2 cells were seeded at a density of 2.0 × 10^5^ cells per well for 24 h. Following pretreatment with a concentration of 20 μM SB203580, a known inhibitor of the protein p38, for 1 h, Gallic acid was subjected to treatment with a concentration of 150 μM. Subsequently, the stability of pro-PARP, p38 and CHOP was evaluated using Western blotting.

### 4.13. Statistical Analysis

The data were presented as means ± standard deviation (SD). Statistically significant differences between the control and treated groups were determined using a Prism *t*-test, with the following significance levels: *, *p* < 0.05; **, *p* < 0.01; ***, and *p* < 0.001, compared to the untreated group.

## 5. Conclusions

The results of the study demonstrated that gallic acid exerted a decrease in cell viability that was dependent on its concentration. Furthermore, it effectively impeded the proliferation of pancreatic cancer cells. The application of gallic acid demonstrated a significant enhancement in ROS concentrations and an upregulation of proteins that are closely linked with ER stress. Furthermore, the activation of p38, a member of the MAPK family, was specifically observed to induce apoptosis. Gallic acid-induced p38 phosphorylation was decreased by NAC’s ability to scavenge reactive oxygen species. The administration of SB203580, a specific inhibitor of p38, decreased the expression of CHOP that the gallic acid induced. This work establishes a correlation between the endoplasmic reticulum stress pathway and the p38 pathway. Gallic acid induces the production of ROS, which then activate the p38 and ER stress signaling pathways. The synergy between the two approaches is evident. This study additionally established a correlation with nitric oxide. According to the findings of this investigation, it has been suggested that gallic acid exhibits potential therapeutic effects in the treatment of pancreatic cancer. These findings present a novel prospect for the advancement of pancreatic cancer investigation and therapy.

## Figures and Tables

**Figure 1 ijms-24-15236-f001:**
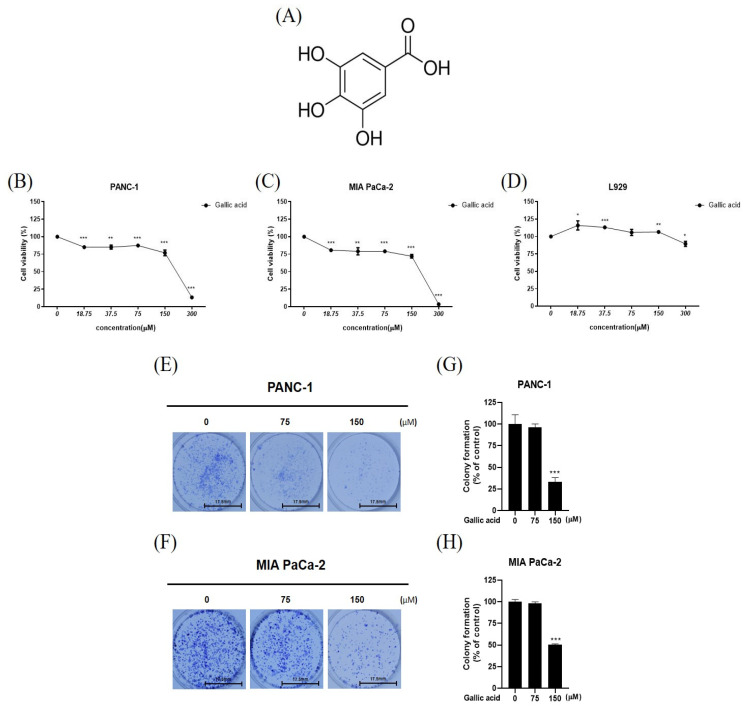
Cytotoxic effect of gallic acid on PANC-1, MIA PaCa-2 and l-929 cells. (**A**) Chemical structure of gallic acid (**B**–**D**) Cell viability after treating PANC-1, MIA PaCa-2 and L-929 cells with various concentrations of gallic acid (18.75, 37.5, 75, 150, 200, 250 and 300 μM) for 24 h. (**E**,**F**) The provided images represent colony formation (on the left) and (**G**,**H**) a bar graph (on the right) illustrating the effects of gallic acid treatment at concentrations of 75 μM and 150 μM on PANC-1 and MIA PaCa-2 cells. The colonies were subjected to staining with Diff Quick Solution and subsequently quantified using the ImageJ software. The aforementioned figures represent the mean values derived from three distinct experimental trials. Scale bar: 17.5 mm. Data represent means ± SD. * *p* < 0.05, ** *p* < 0.01, *** *p* < 0.001 versus untreated group.

**Figure 2 ijms-24-15236-f002:**
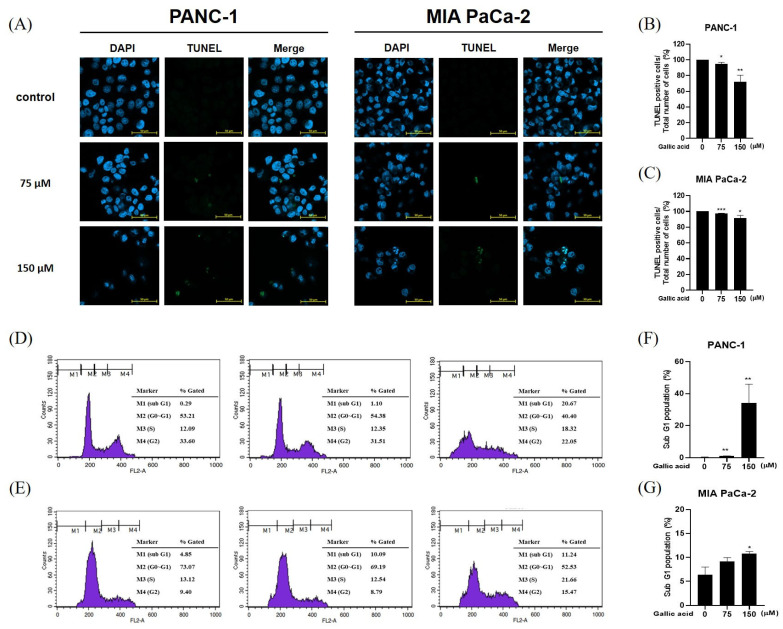
The apoptotic effect of gallic acid in PANC-1 and MIA PaCa-2. (**A**) The occurrence of cell death in PANC-1 and MIA PaCa-2 cell lines was verified by the utilization of a confocal microscope. The nucleus exhibits blue fluorescence at a wavelength of 465 nm, while fluorescent green images are observed in dead cells at a wavelength of 520 nm. (**B**,**C**) The graph depicts the relationship between the total number of cells and the number of cells that exhibit positive staining for the TUNEL assay. Scale bar: 50 μm. (**D**,**E**) The effect of gallic acid (75 μM and 150 μM) on the distribution of the cell cycle in PANC-1 and MIA PaCa-2 cells was assessed using fluorescence-activated cell sorting (FACS). (**F**,**G**) The graph represents the ratio between the duration of M1 (sub G1) phase and M4 (G2) phase in the cell cycle. The aforementioned figures represent the mean values derived from three distinct experimental trials. Data represented means ± SD. * *p* < 0.05, ** *p* < 0.01, *** *p* < 0.001 versus untreated group.

**Figure 3 ijms-24-15236-f003:**
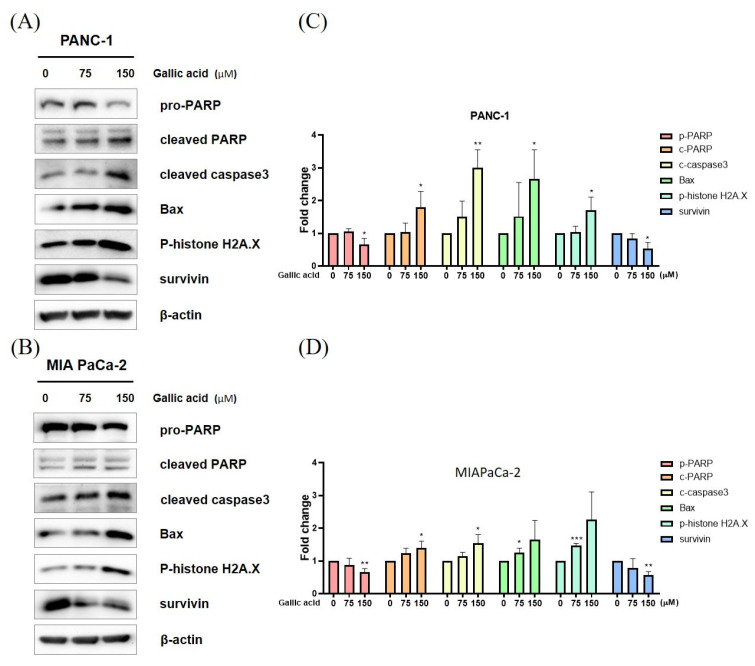
Gallic acid controls proteins involved in apoptosis in pancreatic cancer cells. (**A**,**B**) Protein expression in 75 μM, 150 μM gallic acid-treated PANC-1 and MIA PaCa-2 cells. The protein expression of apoptotic markers, including PARP, caspase3, BAX, p-histone H2A.X, and survivin, was measured by Western blot analysis. (**C**,**D**) The bar graph depicted the proportion of protein expression. The aforementioned figures represent the mean values derived from three distinct experimental trials. Data represented means ± SD. * *p* < 0.05, ** *p* < 0.01, *** *p* < 0.001 versus untreated group.

**Figure 4 ijms-24-15236-f004:**
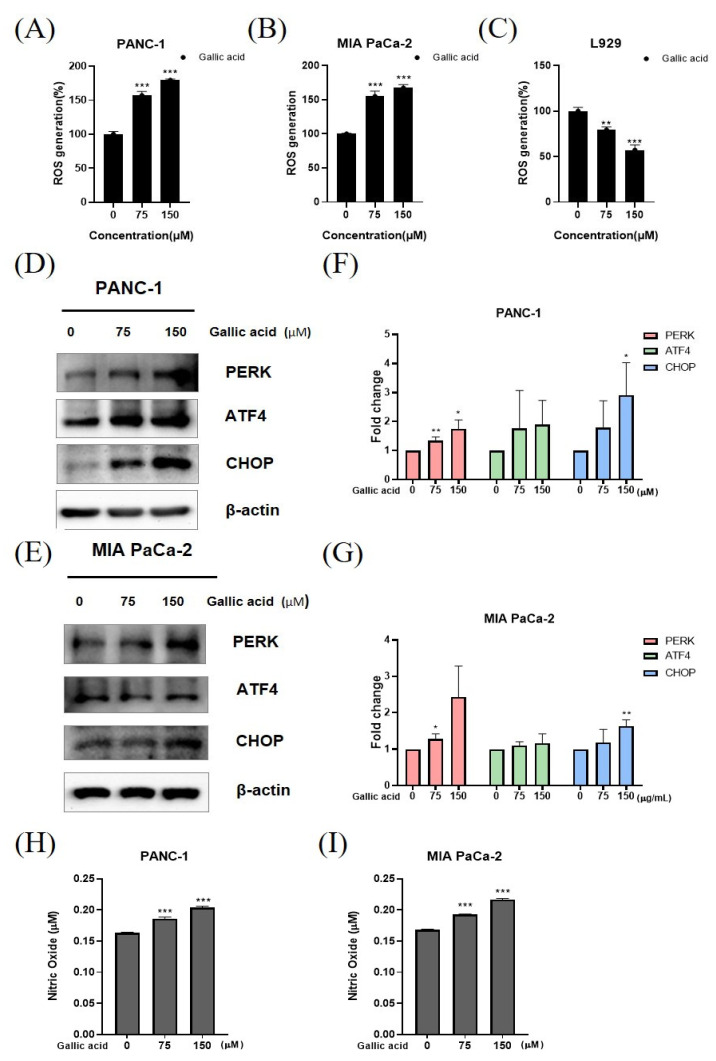
Gallic acid induces ROS generation and apoptosis via ER stress. (**A**–**C**) ROS generation in PANC-1, MIA PaCa-2 and L-929 cells treated with 75 μM, 150 μM gallic acid for 24 h. (**D**,**E**) The effect of gallic acid on ER stress-related apoptosis protein such as PERK, ATF4, and CHOP in PANC-1 and MIA PaCa-2 cells by Western blotting. PANC-1 and MIA PaCa-2 cells were treated with gallic acid (75 μM and 150 μM) for 24 h. (**F**,**G**) The quantification of western blotting was represented using a bar graph. (**H**,**I**) The levels were assessed using the nitric oxide (NO) assay following the treatment with gallic acid. The aforementioned figures represent the mean values derived from three distinct experimental trials. Data represent means ± SD. * *p* < 0.05, ** *p* < 0.01, *** *p* < 0.001 versus untreated group.

**Figure 5 ijms-24-15236-f005:**
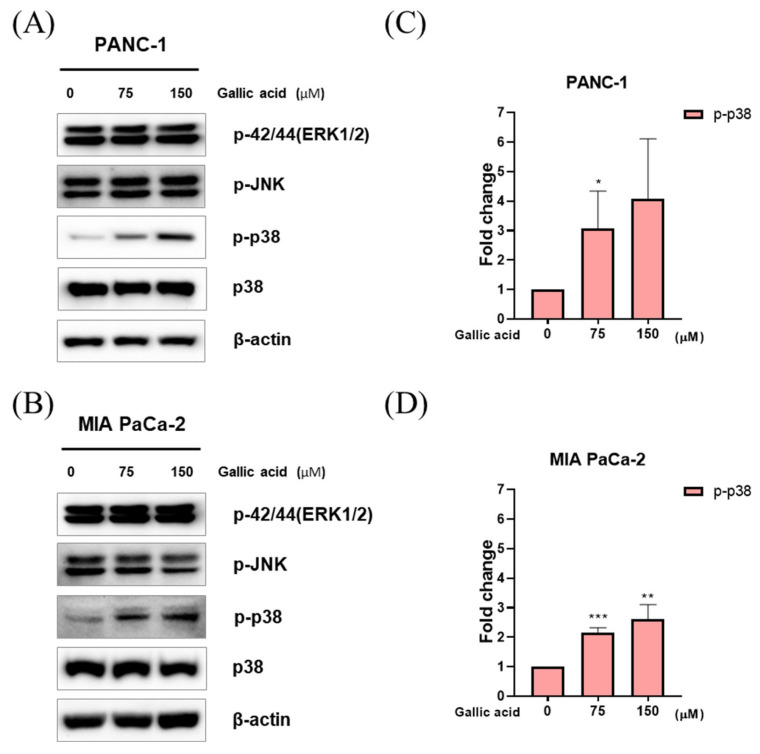
Gallic acid-induced apoptosis through the activation of p38 pathways. (**A**,**B**) MAPK signaling pathway proteins in gallic acid treated PANC-1 and MIA PaCa-2. The p-42/44(ERK1/2), p-JNK, p-p38 and p38 were identified by Western blot analysis. (**C**,**D**) The presented bar graph illustrates the levels of p-p38 expression in PANC-1 and MIA PaCa-2. Data represent means ± SD. * *p* < 0.05, ** *p* < 0.01, *** *p* < 0.001 versus untreated group.

**Figure 6 ijms-24-15236-f006:**
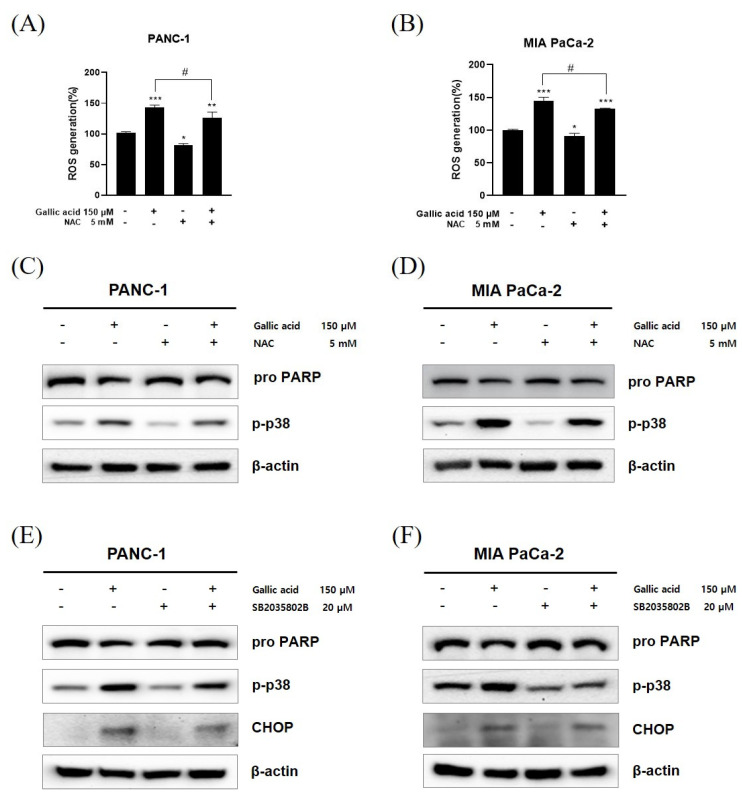
The role of the ROS/p-p38 signal pathway in gallic acid-induced apoptosis. (**A**,**B**) The administration of NAC resulted in a reduction in the production of ROS produced by gallic acid. The alteration in pH resulting from the use of NAC was regulated by the addition of sodium hydroxide specifically intended for cell culture applications. (**C**,**D**) The confirmation of the effect of gallic acid on p38 pathways in NAC-treated PANC-1 and MIA PaCa-2 cells was confirmed. (**E**,**F**) The confirmation of the impact of gallic acid on endoplasmic reticulum stress in PANC-1 and MIA PaCa-2 cells subjected to SB203580 treatment was established. The aforementioned figures represent the mean values derived from three distinct experimental trials. Data represent means ± SD. * *p* < 0.05, ** *p* < 0.01, *** *p* < 0.001 versus untreated group. # *p* < 0.05 versus gallic acid only in the treated group.

**Figure 7 ijms-24-15236-f007:**
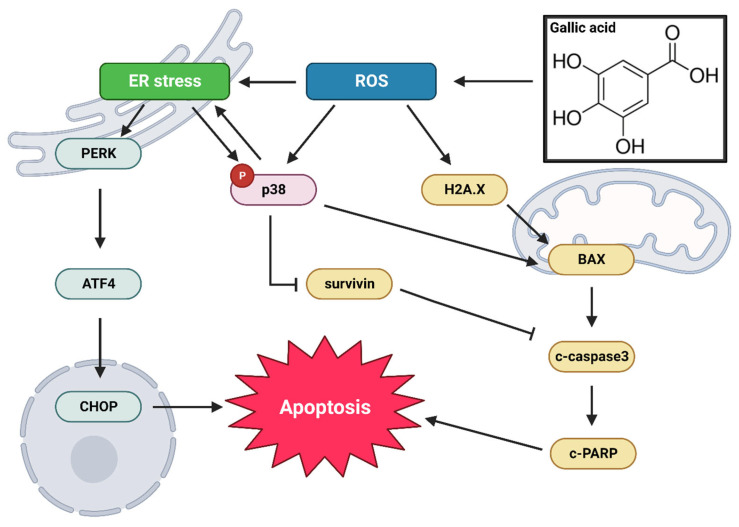
Graphical overview of the effect of gallic acid on pancreatic cancer cell lines PANC-1 and MIA PaCa-2. Gallic acid induces the generation of ROS within biological systems. These ROS, in turn, initiate the activation of two distinct cellular pathways, the p38 pathway and the ER stress pathway. The activation of these pathways subsequently culminates in the process of apoptosis. The two pathways are intricately interconnected and possess the capacity to be simultaneously modulated. The “P” in red indicated the phosphorylation form.

## Data Availability

The data sets generated and/or analyzed in this study are available in the manuscript or can be requested from the author (B.K.) upon reasonable request.

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
