# Peer review of "Apoptotic Effect of Gallic Acid via Regulation of p-p38 and ER Stress in PANC-1 and MIA PaCa-2 Cells Pancreatic Cancer Cells"

_ijms, 2023, doi:10.3390/ijms242015236_

Round 1

Reviewer 1 Report

Review of manuscript ijms-2575219

The manuscript by Kim J.W. and Kim B. “Apoptotic effect of gallic acid via regulation of p-p38 and ER stress in pancreatic cells” is dedicated to an important topic of possible treatments for pancreatic cancer (PC). Authors show the effects of gallic acid (naturally occurring compound in many plants) on pancreatic cell lines. While the decrease of cell viability after treatment with gallic acid (GA) at different concentrations is clear, some other statements need to be examined more deeply. The comments are organised by sections of manuscript.

Manuscript needs a language check, because in many places some words are misused, leading to odd conclusions (see in comments below). Also, in several places the caps lock is not necessary for the words within the sentence (e.g., line 62. ER Stress should be ER stress). I will not point out every single one.

Introduction

Comment 1: One thing missing from the manuscript is the rationale or discussion about chosen PC cell lines. What is similar/different, because in some cases some effects are more pronounced in one but not in both cell lines. This needs to be included either in the Introduction or Discussion part.

Comment 2: line 63 “… ROS-mediated ER stress through the p38 activation pathway can causes (wrong grammar, should be “can cause”) apoptosis [16].” The reference is about cervical cancer cells, and it needs to be mentioned in here as well.

Material and Methods

One main question is – How many repetitions were performed for each experiment? How did the statistical significance was calculated – was it from 3 replicates or triplicates from several (how many) repeated experiments?

Many comments about grammar and style.

In addition, be consequent – if “h” is used for “hours”, please, continue to use “h”, not “hours” randomly switching to “h” and back. The same goes for the tense – if the past tense is used, please, use it throughout.  

Each of following comments is addressed to the subsection.

Comment 3: section 2.1., line 71 “This GA…” – should be “The GA…”. Line 72-73 “GA was dissolved in ethanol and stored in 150 mM stock for the following experiments, then stored at -20C” – could it be better to write “GA stock at 150uM in ethanol was stored at -20C”?

Comment 4: section 2.2., line 81 “Cells were kept in a cell incubator…” – Cells were incubated at 37C, 5% CO2.

Comment 5: section 2.3, line 84 “1.0x104 of…cells were seeded on … plate overnight”. Please, identify if this number of cells is cells/cm2 or cells/well? This comment is also for other sections where cell numbers are mentioned. Secondly, cells are not seeded on plate overnight. They are incubated in media described above (in section 2.1. in manuscript). Cells were seeded at density … cells/cm2 or cells/well and after 24h treated with GA for 24h.

In addition – how was the concentration range chosen for testing GA effect on cell viability?

Comment 6: section 2.4., line 92 “… it was replaced with a new Mmedia…” did authors started to use new, different media or they mean “fresh media”?

How did the authors count the colonies and got the statistical data?

Comment 7: Please, review section 2.5. since there are too many language mistakes to point each. Use of word “treated” vs. “added” and “dyed” vs “stained”.

Comment 8: section 2.5., line 111 “Cells were reacted with RNase A …. In the incubator.” Better “Cells were treated with RNase … at 37C”? One could only imagine the incubator means 37C.

Comment 9: section 2.7., line 116 “…cells were collected. The collected cells were lysed…” Can you make one sentence. Also, how they were collected – by trypsinization? What protein lysis buffer was used?   

Line 120-121 “After collecting the supernatant from cell lysate (unnecessary part - reviewer comment), the same amount of protein was calculated through BCA assay.” The sentence is very confusing – was protein concentration assessed by BCA assay? What is the manufacturer?

Line 123 “…loaded in each well of the SDS-PAGE” – just “loaded on SDS-PAGE” or better “After SDS-PAGE (10-15% at 95V for 110 min) proteins were blotted to nitrocellulose membrane at 290 mA for 120 min.” Please, see all the other sentences in this subsection and re-write accordingly.

Line 134 “...and reacted with secondary antibody…” – did you mean “stained”? Also, was the secondary antibody conjugated with some fluorochrome or other molecule to visualise results? Please, mention it here. What was the dilution for secondary antibody? Did you use the same secondary antibody for all primary antibodies?

Comment 10: section 2.8., lines143 – 144 – At what temperature the staining was made? How long the cells were treated with GA? What concentrations of GA were used? DMEM without phenol red - who is manufacturer? Did you add FBS and pen/strep? While Western blot was described in detail, this section lacks a lot of important information.

Comment 11: section 2.9., line 148 – in addition to previous comments about cell densities (need units) and comment that cells were incubated, not plated overnight, please provide information on how the cells were harvested, was it by trypsinization? Also, a very brief description of the methos could be beneficial.

Comment 12: section 2.10. – for some experiments a bit more calculations and statistical analysis are needed. For example, the TUNEL results are visualized in figure 2.A, but no statistical calculations to see how much more apoptotic cells are in treated group vs untreated group. Since we see lower total cell numbers, it would be necessary to do the normalization.  

Results

Comment 13: section 3.1., line 160 – the first sentences is not needed. Line 162 - GA concentration 300 misses the unit (probably uM). Line 163 – authors state that GA “increased cytotoxicity in a concentration-dependent manner”, however in Figure 1.B and 1.C we can see rather plateau in the range between 18.75 uM and 75 uM. Can you comment on that? How many cell viability measurements were performed? What does the error bars represent – is it STE or STD?

Question! How would you explain that cell viability after GA treatment at 150uM concentration is approx. 75% for both cell lines, but in colony forming assay, we observe not only significant decrease of colonies, but also a significant difference between two cell lines?

What was the rationale not to include higher concentration – looking at cytotoxicity test, IC50 could be approx. 200 uM?

How did the authors make calculations or assessed the number of colonies (for panels F and G in fig.1.? It needs to be included in Materials and Methods.

There is no need to repeat the method part in the Figure legends. It is enough with e.g., “colony formation after 24h GA treatment.”

Comment 14: section 3.2. TUNEL results must be described in more details, including the data visualized in graph, since the Fig.2.A is only representative. Cell cycle analysis usually is done to observe the changes in DNA content. The Figure 2.B. histograms must be improved, marking what is M1, M2 etc. Do they represent different cell cycle phases? The article the authors reference for cell cycle method depicts 2 peaks in which one represents the apoptotic nucleuses and other – normal nucleuses. The labels describing which line is which cell line and which histogram is for which GA concentration are needed.

How did the calculations were made for the graphs (Fig.2.D and E)?

What specific features are for the subG1 cells? Why they are important?

Comment 15: section 3.3. How did the WB pictures were analysed? How did the calculations were made for the graphical representation? How many repetitions of WB experiments were made?

Comment 16: section 3.4. How did authors choose the ER stress markers to test?

Comment 17: section 3.6. The method needs to be described in Materials and Methods section. Line 239 “The relationship between the p38 pathway and ER stress was confirmed by treating NAC...and SB… respectively.” Threating which cells, how many? Did the cells were treated with GA?

Line 240 “Pretreatment with 5uM NAC for 2 h resulted in reduction in ROS production induced by GA.” How did the timing of pretreatment was assessed? Similar questions are about line 244 where the description of SB treatment is described.

ROS reduction in GA treated cells looks the same as in non-treated cells. Could authors comment on this?

Figure 6.C-F – Some of the changes are not so profound and more detailed analysis is needed, perhaps graphs. Looking at WB images in Figure 5 A and B and Figure C – F, one can see that the band intensity of p-p38 in case of 150 uM GA treatment is similar in Fig. 5 A and B, but in Fig. 6 the p-p38 bands are more intense in MIA-PaCa-2 cells. Could authors comment on this?

A quantitative analysis showing NAC and SB203580 effect on control and GA-treated cells is needed for all tested proteins. For example, it is hard to see the reduction of PARP levels in WB images.

Discussion

Comment 18: line 261 – “Agrimoniim confirmed that it promotes…”. Agrimoniim is a substance that was tested, so it cannot confirm its effects.

Comment 19: line 263-266 – the article of reference 19 was not accessible.

Comment 20: line 266-268 – the statement that authors make is not in line with the statement in reference. GA does NOT allow gastritis to convert into gastric cancer.

Comment 21: line 272 – sharp decrease in cell viability above 150 uM means that authors should have performed next cytotoxicity test with smaller concentration window and with different GA exposure times.

Comment 22: line 274 – TUNEL assay analysis must be dome more in details.

Comment 23: line 275 – 276 – “This means that GA suppressed the growth of …cells.” While I agree that the data suggest the growth arrest of these cells, earlier in results authors talk about cell cycle analysis and apoptosis. Authors should carefully re-examine what cell cycle analysis using PI means, because in line 279 authors state that PI staining “enables the labelling of DNA fragmentation resulting from apoptosis” which is not exactly true. The reference 22 explains the PI staining principle very clearly.

Comment 24: line 281-282 – What exactly leads the authors to the statement that increased numbers of sub-G1 phase cells mean that GA cause the apoptosis? Could it be growth arrest? What are the features of subG1 phase cells?

Comment 25: line 286 – 287 – Does the decrease of PARP means that GA leads to DNA repair inhibition?

Comment 26: line 289 – “… is phosphorylated during apoptosis DNA fragmentation.” – apoptosis-induced DNA fragmentation.

Comment 27: line 293 – “Changes in each protein (which protein? – rev.comment) were confirmed through WB, and apoptosis was demonstrated.” – I strongly disagree that apoptosis was demonstrated. This was just an observation. Authors need more proof for such statement.

Comment 28: line 304 – For that statement reference (31) should be more precise. Existing reference 31 is a review article. Here an original paper describing ATF4 and CHOP relationships should be mentioned.

Comment 29: lines 305-306 – “The WB analysis revealed un upregulation of each protein involved in PERK pathway.” – this statement is not based in WB results, since not all proteins were significantly upregulated.

Comment 30: lines 312-313 – this observational study does not produce enough evidence for such big statement. The best, this study results lead to suggestion of association between NO and ER stress.

Comment 31: lines 326 – the whole sentences does not make any sense – “the GA was treated after NAC treatment…” – what do authors mean by this? Cells were treated with GA and NAC? Line 327-328 “As a result, it was confirmed that 5 mM was appropriate concentration.” – appropriate for what? What was the expected outcome authors were looking for?

Line 328-330 – it is worth to mention here as well that NAC is ROS scavenger and SB… is p38 inhibitor. Was the reduction of CHOP significant? There are not enough data for that.

Line 332 -333 – it is the repetition of results, but does it mean that ROS is upstream p38 and regulates the expression of p38?

Comment 32: line 333-334 “…the decrease in p-p38 causes a decrease in proteins involved in ER stress.” – The results show the decrease of ER stress proteins due to GA treatment. The existing data are insufficient for this statement. What are other regulators of ER stress proteins?  

Comment 33: on Figure 6 experiments – the images show changing levels of proteins in presence or absence of GA and NAC and SB; however, these results are not enough to make statements about apoptosis. Did authors measure cell viability in each group, did the authors assessed the apoptosis levels in each group by e.g., TUNE assay?   

While the conclusions are described clearly, some parts, like association between ER stress and p38, needs more evidence.

As stated before, manuscript must be revised by English language specialist. 

Author Response

Review of manuscript ijms-2575219

The manuscript by Kim J.W. and Kim B. “Apoptotic effect of gallic acid via regulation of p-p38 and ER stress in pancreatic cells” is dedicated to an important topic of possible treatments for pancreatic cancer (PC). Authors show the effects of gallic acid (naturally occurring compound in many plants) on pancreatic cell lines. While the decrease of cell viability after treatment with gallic acid (GA) at different concentrations is clear, some other statements need to be examined more deeply. The comments are organised by sections of manuscript.

Manuscript needs a language check, because in many places some words are misused, leading to odd conclusions (see in comments below). Also, in several places the caps lock is not necessary for the words within the sentence (e.g., line 62. ER Stress should be ER stress). I will not point out every single one.

Introduction

Comment 1: One thing missing from the manuscript is the rationale or discussion about chosen PC cell lines. What is similar/different, because in some cases some effects are more pronounced in one but not in both cell lines. This needs to be included either in the Introduction or Discussion part.

-> (Response) First of all, we would like to express our sincere gratitude for the time and effort the reviewer had put into reviewing our manuscript. We have incorporated changes based on the reviewer comments and revised parts are highlighted by Blue color in the entire revised manuscript. We made modifications and incorporated more details regarding the similarities and differences observed between the two cell lines (page 2, line 84-89).

Comment 2: line 63 “… ROS-mediated ER stress through the p38 activation pathway can causes (wrong grammar, should be “can cause”) apoptosis [16].” The reference is about cervical cancer cells, and it needs to be mentioned in here as well.

>> (Response) We modified and added more information (page 2, line 75-78).

Material and Methods

One main question is – How many repetitions were performed for each experiment? How did the statistical significance was calculated – was it from 3 replicates or triplicates from several (how many) repeated experiments?

 -> (Response) Three repeated trials were undertaken. The statistical significance of the PRISM data was assessed through the utilization of a T-Test. The information provided in the figure legend has been changed and expanded.

Many comments about grammar and style.

In addition, be consequent – if “h” is used for “hours”, please, continue to use “h”, not “hours” randomly switching to “h” and back. The same goes for the tense – if the past tense is used, please, use it throughout.  

Each of following comments is addressed to the subsection.

 >> (Response) English language in the paper has been thoroughly edited, incorporating the suggested revisions and addressing the areas of concern. Additionally, we have included additional content as per your instructions. Additional information has been edited and incorporated.

Comment 3: section 2.1., line 71 “This GA…” – should be “The GA…”. Line 72-73 “GA was dissolved in ethanol and stored in 150 mM stock for the following experiments, then stored at -20C” – could it be better to write “GA stock at 150uM in ethanol was stored at -20C”?

>> (Response) Modified (page 2, line 93).

Comment 4: section 2.2., line 81 “Cells were kept in a cell incubator…” – Cells were incubated at 37C, 5% CO2.

>> (Response) Modified (page 2, line 103-104).

Comment 5: section 2.3, line 84 “1.0x104 of…cells were seeded on … plate overnight”. Please, identify if this number of cells is cells/cm2 or cells/well? This comment is also for other sections where cell numbers are mentioned. Secondly, cells are not seeded on plate overnight. They are incubated in media described above (in section 2.1. in manuscript). Cells were seeded at density … cells/cm2 or cells/well and after 24h treated with GA for 24h.

>> (Response) We modified it and added more information (page 3, line 108-111).

Comment 6: section 2.4., line 92 “… it was replaced with a new Mmedia…” did authors started to use new, different media or they mean “fresh media”?

How did the authors count the colonies and got the statistical data?

 >> (Response) We modified it and added more information (page 3, line 116-122).

Comment 7: Please, review section 2.5. since there are too many language mistakes to point each. Use of word “treated” vs. “added” and “dyed” vs “stained”.

 >> (Response) All of the contents were edited (page 3, line 124-138).

Comment 8: section 2.5., line 111 “Cells were reacted with RNase A …. In the incubator.” Better “Cells were treated with RNase … at 37C”? One could only imagine the incubator means 37C.

 >> (Response) Revised (page 3, line 140).

Comment 9: section 2.7., line 116 “…cells were collected. The collected cells were lysed…” Can you make one sentence. Also, how they were collected – by trypsinization? What protein lysis buffer was used?   

 >> (Response) Revised (page 4, line 147-150).

Line 120-121 “After collecting the supernatant from cell lysate (unnecessary part - reviewer comment), the same amount of protein was calculated through BCA assay.” The sentence is very confusing – was protein concentration assessed by BCA assay? What is the manufacturer?

 >> (Response) Revised (page 4, line 152-153).

Line 123 “…loaded in each well of the SDS-PAGE” – just “loaded on SDS-PAGE” or better “After SDS-PAGE (10-15% at 95V for 110 min) proteins were blotted to nitrocellulose membrane at 290 mA for 120 min.” Please, see all the other sentences in this subsection and re-write accordingly.

 >> (Response) Revised  (page 4, line 154-162).

Line 134 “...and reacted with secondary antibody…” – did you mean “stained”? Also, was the secondary antibody conjugated with some fluorochrome or other molecule to visualise results? Please, mention it here. What was the dilution for secondary antibody? Did you use the same secondary antibody for all primary antibodies?

 >> (Response) Revised (page 4, line 167-170).

Comment 10: section 2.8., lines143 – 144 – At what temperature the staining was made? How long the cells were treated with GA? What concentrations of GA were used? DMEM without phenol red - who is manufacturer? Did you add FBS and pen/strep? While Western blot was described in detail, this section lacks a lot of important information.

 >> (Response) We modified it and added information in detail about how ROS was measured (page 4, line 174-177) and what was done after dividing NAC by 2.11 and adding western blotting assay (page 5, line 192-197).

Comment 11: section 2.9., line 148 – in addition to previous comments about cell densities (need units) and comment that cells were incubated, not plated overnight, please provide information on how the cells were harvested, was it by trypsinization? Also, a very brief description of the methos could be beneficial.

 >> (Response) Revised (page 5, line 192-197).

Comment 12: section 2.10. – for some experiments a bit more calculations and statistical analysis are needed. For example, the TUNEL results are visualized in figure 2.A, but no statistical calculations to see how much more apoptotic cells are in treated group vs untreated group. Since we see lower total cell numbers, it would be necessary to do the normalization.  

 >> (Response) We made detailed modifications.

Results

Comment 13: section 3.1., line 160 – the first sentences is not needed. Line 162 - GA concentration 300 misses the unit (probably uM).

 >> (Response) Revised (page 5, line 128).

Line 163 – authors state that GA “increased cytotoxicity in a concentration-dependent manner”, however in Figure 1.B and 1.C we can see rather plateau in the range between 18.75 uM and 75 uM. Can you comment on that? How many cell viability measurements were performed? What does the error bars represent – is it STE or STD?

 >> (Response) We made modifications to all relevant content (page 5, line 213-217).

The statistical significance was assessed by doing a T-Test using the PRISM software, as indicated in the figure legend.

Question! How would you explain that cell viability after GA treatment at 150uM concentration is approx. 75% for both cell lines, but in colony forming assay, we observe not only significant decrease of colonies, but also a significant difference between two cell lines?

>> (Response) For the colony formation assay, cells were incubated for 9 days after exposure to the GA. That is the reason why there are difference between cell viability assay results and colony formation assay. The western blot analysis results showed that the more changed apoptosis protein expression were measured in PANC-1 than MIA PaCa-2 in figure 3.

What was the rationale not to include higher concentration – looking at cytotoxicity test, IC50 could be approx. 200 uM?

 >> (Response) For apoptosis studies, too toxic concentration would be avoided, because it is hard to collect the cell sample for experiments and necrosis could be occurred. 10-30% toxic concentrations are commonly used.

How did the authors make calculations or assessed the number of colonies (for panels F and G in fig.1.? It needs to be included in Materials and Methods.

There is no need to repeat the method part in the Figure legends. It is enough with e.g., “colony formation after 24h GA treatment.”

 >> (Response) The quantification of colonies was conducted using the ImageJ software program. We made modifications in the methods and figure legends section (page 3, line 121-122).

Comment 14: section 3.2. TUNEL results must be described in more details, including the data visualized in graph, since the Fig.2.A is only representative. Cell cycle analysis usually is done to observe the changes in DNA content. The Figure 2.B. histograms must be improved, marking what is M1, M2 etc. Do they represent different cell cycle phases?

 >> (Response) We made modifications to all relevant in figure 2 D, E and legend (page 7, line 252-253).

The article the authors reference for cell cycle method depicts 2 peaks in which one represents the apoptotic nucleuses and other – normal nucleuses. The labels describing which line is which cell line and which histogram is for which GA concentration are needed.

How did the calculations were made for the graphs (Fig.2.D and E)?

>> (Response) The provided graph visually depicts the relationship between the duration of the M1 (sub G1) phase and the M4 (G2) phase within the cell cycle. We added aboded information legend (page 7-8, line 254-256).

What specific features are for the subG1 cells? Why they are important?

>> (Response) Numerous studies have provided strong proof regarding the accumulation of cells in the subG1 phase, well known as the initiation of apoptosis, through the activation of annexin V and caspases 3 [2-8]. These events are demonstrated by comprehensive investigation and persuasive proof. Based on this theory, it may be suggested that subG1 plays a significant role in the process of apoptosis.

Comment 15: section 3.3. How did the WB pictures were analysed? How did the calculations were made for the graphical representation? How many repetitions of WB experiments were made?

>> (Response) The experiment was performed in triplicate, and the assessment of statistical significance was conducted utilizing the software tool, Image J.

Comment 16: section 3.4. How did authors choose the ER stress markers to test?

>> (Response) It is widely acknowledged that the administration of chemotherapy triggers a process known as apoptosis, in various cancer cells. This response is mediated by the release of intracellular ROS. Henceforth, the administration of pharmaceutical intervention resulted in an upregulation of various indicators associated with the presence of ER stress. These markers encompass p-PERK, p-eIF2α, ATF4, and CHOP[9-11].

Comment 17: section 3.6. The method needs to be described in Materials and Methods section.

 >> (Response) We modified and added more information in Materials and Methods section (page 4, line 174-177) and (page 5, line 192-197).

Line 239 “The relationship between the p38 pathway and ER stress was confirmed by treating NAC...and SB… respectively.” Threating which cells, how many? Did the cells were treated with GA?

 >> (Response) We have made alterations to the all aspects of the contents provided in section 3.6 (page 11, line 308-318).

.

Line 240 “Pretreatment with 5uM NAC for 2 h resulted in reduction in ROS production induced by GA.” How did the timing of pretreatment was assessed? Similar questions are about line 244 where the description of SB treatment is described.

 >> (Response) We modified and added more information in Materials and Methods section (page 4, line 174-177) and (page 5, line 192-197).

ROS reduction in GA treated cells looks the same as in non-treated cells. Could authors comment on this?

>> (Response) Upon careful review, it is evident that there exists a marginal diminution in the scale of the observed measurement.

Figure 6.C-F – Some of the changes are not so profound and more detailed analysis is needed, perhaps graphs. Looking at WB images in Figure 5 A and B and Figure C – F, one can see that the band intensity of p-p38 in case of 150 uM GA treatment is similar in Fig. 5 A and B, but in Fig. 6 the p-p38 bands are more intense in MIA-PaCa-2 cells. Could authors comment on this?

>> (Response) The time of exposure varies when capturing the band using the equipment in question. When examining the process of Figure 5, it is observed that the majority of instances exhibit favourable outcomes when the membranes are juxtaposed. Consequently, these instances are characterized by their brevity. Conversely, in the case of Figure 6, it is noted that the outcomes are generally unfavourable, as the membranes fail to align effectively. The observed outcome exhibited variation due to the prolonged length of retention. Nevertheless, it is important to note that all experimental conditions, including the protein the amount and the dilution ratio of the primary and secondary antibody, remained consistent throughout the study.

A quantitative analysis showing NAC and SB203580 effect on control and GA-treated cells is needed for all tested proteins. For example, it is hard to see the reduction of PARP levels in WB images.

>> (Response) The most important focus of our study lies in elucidating the intricate interplay between ROS and an unidentified entity known as p-p38. We intended to direct focus toward the primary objective and present the results demonstratively. The study's primary objective was to determine the effect of GA through the inhibition of ROS (NAC) and p-p38 (SB20358), so no consideration was given to the remaining variables.

Discussion

Comment 18: line 261 – “Agrimoniim confirmed that it promotes…”. Agrimoniim is a substance that was tested, so it cannot confirm its effects.

>> (Response) We modified contents (page 12, line 338-341).

Comment 19: line 263-266 – the article of reference 19 was not accessible.

>> (Response) The identification PMID: 34755289 corresponds to reference 21.

Comment 20: line 266-268 – the statement that authors make is not in line with the statement in reference. GA does NOT allow gastritis to convert into gastric cancer.

>> (Response) As previously stated, it has been demonstrated that GA possesses the ability to inhibit diverse cellular mechanisms implicated in the onset, advancement, and advancement of cancer in numerous malignancies.

Comment 21: line 272 – sharp decrease in cell viability above 150 uM means that authors should have performed next cytotoxicity test with smaller concentration window and with different GA exposure times.

>> (Response) According to the instructions, we included the experimental results part as Figure 1, D in the section (page 6).

Comment 22: line 274 – TUNEL assay analysis must be dome more in details.

 >> (Response) We made modifications to all relevant in figure 2 D, E and legend (page 7, line 252-253).

Comment 23: line 275 – 276 – “This means that GA suppressed the growth of …cells.” While I agree that the data suggest the growth arrest of these cells, earlier in results authors talk about cell cycle analysis and apoptosis. Authors should carefully re-examine what cell cycle analysis using PI means, because in line 279 authors state that PI staining “enables the labelling of DNA fragmentation resulting from apoptosis” which is not exactly true. The reference 22 explains the PI staining principle very clearly.

 >> (Response) The elucidation of the answer was elaborated upon extensively in the revised segment. (page 12, line 358-370).

.

Comment 24: line 281-282 – What exactly leads the authors to the statement that increased numbers of sub-G1 phase cells mean that GA cause the apoptosis? Could it be growth arrest? What are the features of subG1 phase cells?

>> (Response) Numerous studies have provided strong proof regarding the accumulation of cells in the subG1 phase, well known as the initiation of apoptosis, through the activation of annexin V and caspases 3 [2-8]. These events are demonstrated by comprehensive investigation and persuasive proof. Based on this theory, it may be suggested that subG1 plays a significant role in the process of apoptosis.

Comment 25: line 286 – 287 – Does the decrease of PARP means that GA leads to DNA repair inhibition?

 ->(Response) The answer was explained in detail in the revised part of the paper, so we changed.

Comment 26: line 289 – “… is phosphorylated during apoptosis DNA fragmentation.” – apoptosis-induced DNA fragmentation.

->(Response) The answer was explained in detail in the revised part of the paper (page 13, line 382-390).

Comment 27: line 293 – “Changes in each protein (which protein? – rev.comment) were confirmed through WB, and apoptosis was demonstrated.” – I strongly disagree that apoptosis was demonstrated. This was just an observation. Authors need more proof for such statement.

->(Response) The answer was further elaborated upon within the extensively revised section of the paper (page 12-13, line 373-401).

Comment 28: line 304 – For that statement reference (31) should be more precise. Existing reference 31 is a review article. Here an original paper describing ATF4 and CHOP relationships should be mentioned.

-> (Response) It is widely acknowledged that the administration of chemotherapy triggers a process known as programmed cell death, or apoptosis, in various cancer cells. This response is mediated by the release of intracellular reactive oxygen species (ROS). Henceforth, the administration of pharmaceutical intervention resulted in an upregulation of various indicators associated with the presence of endoplasmic reticulum (ER) stress. These markers encompass p-PERK, p-eIF2α, ATF4, and CHOP[9-11].

Comment 29: lines 305-306 – “The WB analysis revealed un upregulation of each protein involved in PERK pathway.” – this statement is not based in WB results, since not all proteins were significantly upregulated.

-> (Response) The majority of the comparatively positive variations in expression for variable c appear to have a positive correlation with concentration. However, following the provided advice, I substituted it with an alternative that is more visible (page 9, Figure 4 E).

Comment 30: lines 312-313 – this observational study does not produce enough evidence for such big statement. The best, this study results lead to suggestion of association between NO and ER stress.

>> (Response) The main point of our study is the process of regulating ER stress by ROS and the synergistic effect of regulating NO and ROS synergistically on preventing cancer. Since NO is about another part of the immune system, we are considering about expanding on to a different part of this study.

Comment 31: lines 326 – the whole sentences does not make any sense – “the GA was treated after NAC treatment…” – what do authors mean by this? Cells were treated with GA and NAC? Line 327-328 “As a result, it was confirmed that 5 mM was appropriate concentration.” – appropriate for what? What was the expected outcome authors were looking for?

 >> (Response) We modified and added more information in Materials and Methods section (page 4, line 174-177) and (page 5, line 192-197).

Line 328-330 – it is worth to mention here as well that NAC is ROS scavenger and SB… is p38 inhibitor. Was the reduction of CHOP significant? There are not enough data for that.

>> (Response) When comparing the relative expression levels, it is observed that the group treated just with gallic acid exhibits a significantly higher expression amount compared to the group treated with both gallic acid and SB2035802b concurrently. Additionally, it is worth noting that the number of groups subjected to the combined treatment is considerably lower. Hence, it is evident that the difference in CHOP quantity beyond that observed in the band.

Line 332 -333 – it is the repetition of results, but does it mean that ROS is upstream p38 and regulates the expression of p38?

>> (Response) The elucidation of the answer was simply elaborated around within a revised section of the manuscript (page 13, line 409-422).

.

Comment 32: line 333-334 “…the decrease in p-p38 causes a decrease in proteins involved in ER stress.” – The results show the decrease of ER stress proteins due to GA treatment. The existing data are insufficient for this statement. What are other regulators of ER stress proteins?  

>> (Response) The elucidation of the answer was simply elaborated around within a revised section of the manuscript (page 13, line 449-452).

Comment 33: on Figure 6 experiments – the images show changing levels of proteins in presence or absence of GA and NAC and SB; however, these results are not enough to make statements about apoptosis. Did authors measure cell viability in each group, did the authors assessed the apoptosis levels in each group by e.g., TUNE assay?   

>> (Response) The elucidation of the answer was simply elaborated around within a revised section of the manuscript (page 14, line 449-459).

While the conclusions are described clearly, some parts, like association between ER stress and p38, needs more evidence.

>> (Response) The elucidation of the answer was simply elaborated around within a revised section of the manuscript (page 13, line 402-409, line 425-432).

  1. Jakštys, B., et al., Different Cell Viability Assays Reveal Inconsistent Results After Bleomycin Electrotransfer In Vitro. J Membr Biol, 2015. 248(5): p. 857-63.
  2. Lin, Z.H., et al., ZnF3, a sulfated polysaccharide from Antrodia cinnamomea, inhibits lung cancer cells via induction of apoptosis and activation of M1-like macrophage-induced cell death. Int J Biol Macromol, 2023. 238: p. 124144.
  3. Heiat, M., et al., Knockdown of SIX4 inhibits pancreatic cancer cells via apoptosis induction. Med Oncol, 2023. 40(10): p. 287.
  4. Loomis, R.E. and J.L. Alderfer, Halogenated nucleic acids: effects of 5-fluorouracil on the conformation and properties of a polyribonucleotide and its constituents. Biopolymers, 1986. 25(4): p. 571-600.
  5. Lee, J.C., et al., Auraptene Induces Apoptosis via Myeloid Cell Leukemia 1-Mediated Activation of Caspases in PC3 and DU145 Prostate Cancer Cells. Phytother Res, 2017. 31(6): p. 891-898.
  6. Huang, H., et al., Papaverine selectively inhibits human prostate cancer cell (PC-3) growth by inducing mitochondrial mediated apoptosis, cell cycle arrest and downregulation of NF-κB/PI3K/Akt signalling pathway. J buon, 2017. 22(1): p. 112-118.
  7. Han, B., J. Wu, and L. Huang, Induction of Apoptosis in Lung Cancer Cells by Viburnum grandiflorum via Mitochondrial Pathway. Med Sci Monit, 2020. 26: p. e920265.
  8. Marcinkute, M., et al., Fluoxetine selectively induces p53-independent apoptosis in human colorectal cancer cells. Eur J Pharmacol, 2019. 857: p. 172441.
  9. Kim, T.W. and S.G. Ko, The Herbal Formula JI017 Induces ER Stress via Nox4 in Breast Cancer Cells. Antioxidants (Basel), 2021. 10(12).
  10. Ma, B., et al., Corosolic acid, a natural triterpenoid, induces ER stress-dependent apoptosis in human castration resistant prostate cancer cells via activation of IRE-1/JNK, PERK/CHOP and TRIB3. J Exp Clin Cancer Res, 2018. 37(1): p. 210.
  11. Xie, W.Y., et al., Acid-induced autophagy protects human lung cancer cells from apoptosis by activating ER stress. Exp Cell Res, 2015. 339(2): p. 270-9.

Reviewer 2 Report

Pancreatic carcinoma (PC) is a major cancerous disease with an extremely unfavorable 5-year survival rate of less than 10%. The typical treatment regimen is based on surgery as the initial treatment, followed by adjuvant chemotherapy. Unfortunately, the four common cytostatics are only effective in 30% of patients after metastases have occurred. A primary goal in the development of new therapies for PC is therefore the development or identification of new, more effective cytostatic substances. In the present study, the authors examined the effects of the potential anti-cancer agent gallic acid, a naturally compound found in a variety of herbal products, on two pancreatic cancer cell lines (PANC-1 and MIA PaCa-2). Nitric oxide, ROS, cell viability, cell cycle and TUNEL assays, as well as Western blot analyzes of selected apoptosis and ER stress proteins were performed. The experiments showed that gallic acid effectively increased ROS levels and upregulated proteins associated with ER stress. Gallic acid activates both the p38 pathway and the ER stress pathway by generating ROS that can induce apoptosis in PC cell lines. These findings provide a new avenue for future research and potential treatment strategies for PC cancer.

The conception of the paper is conclusive. The experimental designs are adequate. The manuscript is in generally well written.

The following things struck me while reading the manuscript:

• Page 1, line 32: Do the values for "death" and "cases" refer to worldwide data?

• Page 2, line 65: "Gallic acid has been used in a variety of therapeutic agents."
Please add literature for this statement.

• In the methods section, please not only state the kit name, but also briefly mention the measuring principle, e.g. Griess assay or MTS.

• In my opinion, the concentrations for the viability assay are not optimal. In the concentration range of 18.75 to 150 µM gallic acid, the viability values are almost unchanged at 75% to 80%. The viability only decreases significantly between 150 and 300 µM. In order to get a clearer picture of the concentration dependence of the viability on the gallic acid, it would be desirable to test at least two other concentrations between 150 and 300 µM. Please complete this data.

• Page 13, line 313: "..but also suggests the novel possibility of an association between NO and ER stress.": This fact is not new; it was already mentioned by Nakato et al (2015) in Scientific Reports (DOI https://doi .org/10.1038/srep14812). Please change the Discussion section accordingly.

Author Response

Comments and Suggestions for Authors

Pancreatic carcinoma (PC) is a major cancerous disease with an extremely unfavorable 5-year survival rate of less than 10%. The typical treatment regimen is based on surgery as the initial treatment, followed by adjuvant chemotherapy. Unfortunately, the four common cytostatics are only effective in 30% of patients after metastases have occurred. A primary goal in the development of new therapies for PC is therefore the development or identification of new, more effective cytostatic substances. In the present study, the authors examined the effects of the potential anti-cancer agent gallic acid, a naturally compound found in a variety of herbal products, on two pancreatic cancer cell lines (PANC-1 and MIA PaCa-2). Nitric oxide, ROS, cell viability, cell cycle and TUNEL assays, as well as Western blot analyzes of selected apoptosis and ER stress proteins were performed. The experiments showed that gallic acid effectively increased ROS levels and upregulated proteins associated with ER stress. Gallic acid activates both the p38 pathway and the ER stress pathway by generating ROS that can induce apoptosis in PC cell lines. These findings provide a new avenue for future research and potential treatment strategies for PC cancer.

 -> (Response) First of all, we would like to express our sincere gratitude for the time and effort the reviewer had put into reviewing our manuscript. We have incorporated changes based on the reviewer comments provided in the manuscript which revised parts are highlighted by BLUE color in the entire revised manuscript.

The conception of the paper is conclusive. The experimental designs are adequate. The manuscript is in generally well written.

The following things struck me while reading the manuscript:

  • Page 1, line 32: Do the values for "death" and "cases" refer to worldwide data?

 -> (Response) The reference GLOBOCAN serves as a valuable resource for obtaining global cancer statistics in the year 2020.

  • Page 2, line 65: "Gallic acid has been used in a variety of therapeutic agents."
    Please add literature for this statement.

 >> (Response) We added more reference [17-23] (page 2, line 80).

  • In the methods section, please not only state the kit name, but also briefly mention the measuring principle, e.g. Griess assay or MTS.

 >> (Response) The whole of the content related to the materials and methods section has been revised. (page 3-5, line 99-203).

  • In my opinion, the concentrations for the viability assay are not optimal.In the concentration range of 18.75 to 150 µM gallic acid, the viability values are almost unchanged at 75% to 80%.The viability only decreases significantly between 150 and 300 µM. In order to get a clearer picture of the concentration dependence of the viability on the gallic acid, it would be desirable to test at least two other concentrations between 150 and 300 µM. Please complete this data.

>> (Response) According to the instructions, we included the experimental results part as Figure 1, D in the section (page 6).

Reviewer 3 Report

The authors from this manuscript aimed to investigate the induction of apoptosis by gallic acid and elucidate its anticancer mechanisms of action in the PC cell lines, PANC-66 1 and MIA PaCa-2. The authors demonstrated that gallic acid exhibited a concentration-dependent decrease in cell viability and inhibited cell growth in pancreatic cancer cells. Furthermore, gallic acid effectively increased ROS and NO levels and upregulated proteins associated with ER stress. Additionally, NAC (ROS scavenger) and SB203580 (p38 inhibitor) reversed the level of p-p38, and level of CHOP, which increased due to gallic acid. Overall, the manuscript is well-written and several comments are indicated below:

1. Gallic acid is a well-known polyphenolic compound, which is shown to exhibit potent antioxidant activity, which is in contrary to the data presented in this paper. Is this a specific effect in cancer cell lines at the tested concentration?

2. What is the effect of gallic acid in normal pancreatic cells, in terms of cell viability, ROS levels etc?

3. Gallic acid is shown to increase NO levels in the cancer cells. What is the proposed mechanism for the increase in NO levels? Does gallic acid lead to an increase in iNOS or other NOS, which could potentially lead to the reported increased in NO levels?

4. What is the contribution of gallic acid-induced NO towards cell viability and apoptosis? Additional experiments using L-NAME or specific NOS inhibitor (if it is indeed iNOS) would provide some insights into the mechanisms regarding NO.

N/A

Author Response

The authors from this manuscript aimed to investigate the induction of apoptosis by gallic acid and elucidate its anticancer mechanisms of action in the PC cell lines, PANC-66 1 and MIA PaCa-2. The authors demonstrated that gallic acid exhibited a concentration-dependent decrease in cell viability and inhibited cell growth in pancreatic cancer cells. Furthermore, gallic acid effectively increased ROS and NO levels and upregulated proteins associated with ER stress. Additionally, NAC (ROS scavenger) and SB203580 (p38 inhibitor) reversed the level of p-p38, and level of CHOP, which increased due to gallic acid. Overall, the manuscript is well-written and several comments are indicated below:

-> (Response) First of all, we would like to express our sincere gratitude for the time and effort the reviewer had put into reviewing our manuscript. We have incorporated changes based on the reviewer comments provided in the manuscript which revised parts are highlighted by Blue color in the entire revised manuscript.

  1. Gallic acid is a well-known polyphenolic compound, which is shown to exhibit potent antioxidant activity, which is in contrary to the data presented in this paper. Is this a specific effect in cancer cell lines at the tested concentration?

>> (Response) We modified and added more detailed information [65-69] (page 13, line 411-412).

  1. What is the effect of gallic acid in normal pancreatic cells, in terms of cell viability, ROS levels etc?

>> (Response) According to the instructions, we included the experimental results part as Figure 1, D in the section (page 6).

.

Round 2

Reviewer 1 Report

Review of manuscript ijms-2575219

This is the second round of the review of the manuscript by Kim J.W. and Kim B. “Apoptotic effect of gallic acid via regulation of p-p38 and ER stress in pancreatic cells”.

I thank the authors for responses and improvement of manuscript with previously missing information. However, language check and scientific writing style must be improved.

While I agree on the importance of the topic there are some scientific questions I’d like to address.

1)     Looking at the results altogether (partially combined in Table below), we see that cell viability at 150 uM GA for PANC1 cell line is approx. 80 %, while for MIA PaCa-2 cell line it is approx. 55 %, leading to first conclusion that GA could be more effective in MIA PaCa-2 cell line. Further, looking at one of the indirect methods for apoptosis detection (TUNEL) we see that while GA had better effect (in terms of decreasing cancer cell numbers) in MIA PaCa-2, the apoptotic cells are only approx. 7 % of all the left cells.

Questions to authors: a) why have not Annexin V staining performed for assessment of apoptosis level in cells? b) Would you consider an effect of GA being therapeutically potential if it drives only 30 % of cell population towards apoptosis? Perhaps different time points and some other measurements are needed to justify this potential (that I do not object – just need more evidence and some more direct proof of apoptosis).

PANC1 (after 24h of 150uM GA)

MIA PaCa-2 (after 24h of 150uM GA)

Cell viability – number of viable cells (% of control) (Fig 1.B and C)

Approx. 80 %

Approx. 55 %

Cell proliferation – number of colonies (% of control), (colony forming assay, Fig.1.EG and FH)

Approx. 35 %

Approx. 48 %

% of apoptotic cells (Fig.2.B and C)

Approx. 30 %

Approx. 7 %

% of cells in subG1 phase (Fig.2. D and E)

Increased by 20 %

Increased by 6.39 %

% of cells in S phase (Fig.2. D and E)

Increased by 6.23 %

Increased by 8.54 %

Fold change of cPARP (Fig. 3. C and D)

Increased by 0.6 fold

Increased by 0.4 fold

Fold change of c-caspase3 (Fig. 3. C and D)

Increased by 2 fold

Increased by 0.5 fold

Fold change of BAX (Fig. 3. C and D)

Increased by 1.8 fold

Increased by 0.7 fold

Fold change of survivin (Fig. 3. C and D)

Decreased by 0.3 fold

Decreased by 0.4 fold

ROS generation (Fig.4.A and B

Increased by approx. 80 %

Increased by approx. 65 %

etc.

Discussion lacks such a general overview of all these measurements (not necessarily a table).

2)     Authors use the term “apoptotic mechanism” (e.g. line 351 “… we examined the apoptotic mechanism of gallic acid and its relationship with ROS-mediated p38 activation signaling.”, however, due to many indirect apoptosis measurements (subG1 population increase, elevated c-Caspase 3 and BAX expression),  one could argue the study examines the relationship between ROS-mediated p38 activation with effects of GA, that leads either to slower cell proliferation (shown in Fig.1.EG and FH) or some cell death in PC cells lines. In MIA PaCa-2 cells had more increased cell population in S phase! Could authors provide some explanation?

While there are a lot of data and interesting results, discussion needs to analyse results together, not separately, giving an overview (somewhat given at the end of discussion). Authors do not need to add, but rewrite and re-anlyse the data and perhaps some other conclusions will arise. 

The English language must be improved and style of scientific language as well. 

There are some confusing statements, e.g. Line 416 - 417: "The research study revealed the NO has inhibitory effect on apoptosis [meaning that more NO will lead to LESS apoptotic cells - rev.comment]. While these cells underwent apoptosis when exposed to high quantities of NO [66] [in this sentence authors contradict the previous sentence - rev.comment]."

Author Response

This is the second round of the review of the manuscript by Kim J.W. and Kim B. “Apoptotic effect of gallic acid via regulation of p-p38 and ER stress in pancreatic cells”.

I thank the authors for responses and improvement of manuscript with previously missing information. However, language check and scientific writing style must be improved.

While I agree on the importance of the topic there are some scientific questions I’d like to address.

1)     Looking at the results altogether (partially combined in Table below), we see that cell viability at 150 uM GA for PANC1 cell line is approx. 80 %, while for MIA PaCa-2 cell line it is approx. 55 %, leading to first conclusion that GA could be more effective in MIA PaCa-2 cell line. Further, looking at one of the indirect methods for apoptosis detection (TUNEL) we see that while GA had better effect (in terms of decreasing cancer cell numbers) in MIA PaCa-2, the apoptotic cells are only approx. 7 % of all the left cells.

Questions to authors:

  1. a) why have not Annexin V staining performed for assessment of apoptosis level in cells?

>> (Response) Thank you for your valuable comments. I understand your suggestion but we couldn’t perform the Annexin V because our FACs machine is out order during this study, so we couldn’t do so. However, we performed the TUNEL assay, cell cycle assay and apoptosis related protein analysis by Western blot analysis.

  1. b) Would you consider an effect of GA being therapeutically potential if it drives only 30 % of cell population towards apoptosis? Perhaps different time points and some other measurements are needed to justify this potential (that I do not object – just need more evidence and some more direct proof of apoptosis).

PANC1 (after 24h of 150uM GA)

MIA PaCa-2 (after 24h of 150uM GA)

Cell viability – number of viable cells (% of control) (Fig 1.B and C)

Approx. 80 %

Approx. 55 %

Cell proliferation – number of colonies (% of control), (colony forming assay, Fig.1.EG and FH)

Approx. 35 %

Approx. 48 %

% of apoptotic cells (Fig.2.B and C)

Approx. 30 %

Approx. 7 %

% of cells in subG1 phase (Fig.2. D and E)

Increased by 20 %

Increased by 6.39 %

% of cells in S phase (Fig.2. D and E)

Increased by 6.23 %

Increased by 8.54 %

Fold change of cPARP (Fig. 3. C and D)

Increased by 0.6 fold

Increased by 0.4 fold

Fold change of c-caspase3 (Fig. 3. C and D)

Increased by 2 fold

Increased by 0.5 fold

Fold change of BAX (Fig. 3. C and D)

Increased by 1.8 fold

Increased by 0.7 fold

Fold change of survivin (Fig. 3. C and D)

Decreased by 0.3 fold

Decreased by 0.4 fold

ROS generation (Fig.4.A and B

Increased by approx. 80 %

Increased by approx. 65 %

etc.

Discussion lacks such a general overview of all these measurements (not necessarily a table).

 >> (Response) Thank you for your valuable comments. As you can see in figure 1B and C, GA showed toxic effect up to 80-90% in 200 uM concentration in both cancer cells. However, we decided the experiment concentration which has 10 to 40% toxic effect. That is because, if we use the too toxic concentration, there could be necrosis and we can’t get the enough cells to perform mechanism studies and others. It is very usual selection of concentration in apoptosis studies. Please check the cited studies below.

About the apoptosis, we performed TUNEL assay, and sub G1 population was measured, apoptotic proteins including PARP, caspase-3, p-hitone H2a.x, CHOP, etc. were checked.

Scientific Reports volume 12, Article number: 17729 (2022) 

Phytother Res. 2023 Sep;37(9):4224-4235.  doi: 10.1002/ptr.7903. Epub 2023 May 26.

2)     Authors use the term “apoptotic mechanism” (e.g. line 351 “… we examined the apoptotic mechanism of gallic acid and its relationship with ROS-mediated p38 activation signaling.”, however, due to many indirect apoptosis measurements (subG1 population increase, elevated c-Caspase 3 and BAX expression),  one could argue the study examines the relationship between ROS-mediated p38 activation with effects of GA, that leads either to slower cell proliferation (shown in Fig.1.EG and FH) or some cell death in PC cells lines. In MIA PaCa-2 cells had more increased cell population in S phase! Could authors provide some explanation?

 >> (Response) First of all, TUENL assay and cell cycle assay results are the direct apoptosis experiments. Also, S phase cell populations were increased. After cell cycle arrest, the cells undergo apoptosis.

While there are a lot of data and interesting results, discussion needs to analyse results together, not separately, giving an overview (somewhat given at the end of discussion). Authors do not need to add, but rewrite and re-anlyse the data and perhaps some other conclusions will arise. 

  >> (Response) Thank you for your comments. Revised.

The English language must be improved and style of scientific language as well. 

There are some confusing statements, e.g. Line 416 - 417: "The research study revealed the NO has inhibitory effect on apoptosis [meaning that more NO will lead to LESS apoptotic cells - rev.comment]. While these cells underwent apoptosis when exposed to high quantities of NO [66] [in this sentence authors contradict the previous sentence - rev.comment]."
 >> (Response) We made modifications (page 13, line 417-418).

  1. Jakštys, B., et al., Different Cell Viability Assays Reveal Inconsistent Results After Bleomycin Electrotransfer In Vitro. J Membr Biol, 2015. 248(5): p. 857-63.

Round 3

Reviewer 1 Report

This is the third round of the review of the manuscript by Kim J.W., et al., “Apoptotic effect of gallic acid via regulation of p-p38 and ER stress in pancreatic cells”.

I thank the authors for responses and improvement of manuscript - at some point with information that is not necessary (e.g. Line 242, information could be transffered to discussion). Our discussion about methods to prove apoptosis does not have to be reflected in the manuscript and there is no reason to be offended by questions.

Suggestions and comments:

Title – just to be precise and clear, it would be nice to point that the study is in PC cell LINES.

Abstract – please, see Line 34 – I strongly believe this study is about pancreatic, not prostate cancer!

Introduction – Line 85 – 87: author’s state that they “… illustrate the early morphological changes in the peritoneal metastasis”, however the main changes authors illustrate are molecular. In Line 86 authors state “… the connection between these morphological dynamics and their oncogenic properties was examined” – while I agree tha study shows molecular mechanism affected by GA, which experiments would subtantiate this statement? Which oncogenic properties were examined in this study?

Materials and Methods – perhaps it is a type-o in Line 93 that GA stock at 150 uM in EtOH was made, because it would be hard to use it to make 300 uM dilution for cell viability assay. Was it 150 mM? Another type-o is in Line 112 – double 300 uM .

Wording – please, choose one – cells planted or cells seeded (this would be better) and use it everywhere.

Since GA was diluted in ethanol, did authors use a solvent control in the experiments?

Results -  

Rev comment: a) why have not Annexin V staining performed for assessment of apoptosis level in cells?

Author’s response:  Thank you for your valuable comments. I understand your suggestion but we couldn’t perform the Annexin V because our FACs machine is out order during this study, so we couldn’t do so. However, we performed the TUNEL assay, cell cycle assay and apoptosis related protein analysis by Western blot analysis.

Rev response: I believe, the data for Figure 2.D.E. were obtained using FACS. Also, Annexin staining can be read by IF microscopy as well. However, I would like to close this part of our discussion. As pointed many times by you – yes, WB and differences in caspases and other apoptosis-related proteins are detected.

Discussion –

Rev comment: There are some confusing statements, e.g. Line 416 - 417: "The research study revealed the NO has inhibitory effect on apoptosis [meaning that more NO will lead to LESS apoptotic cells - rev.comment]. While these cells underwent apoptosis when exposed to high quantities of NO [66] [in this sentence authors contradict the previous sentence - rev.comment]."

Author’s response:  We made modifications (page 13, line 417-418). 

  1. Jakštys, B., et al., Different Cell Viability Assays Reveal Inconsistent Results After Bleomycin Electrotransfer In Vitro. J Membr Biol, 2015. 248(5): p. 857-63.

Rev response: Please, read the paragraph – those 2 sentences (marked red in author’s revised manuscript) could be integrated better.  

The language is improved, but some last editing could help (see comments of type-o etc). 

Author Response

Comments and Suggestions for Authors

This is the third round of the review of the manuscript by Kim J.W., et al., “Apoptotic effect of gallic acid via regulation of p-p38 and ER stress in pancreatic cells”.

I thank the authors for responses and improvement of manuscript - at some point with information that is not necessary (e.g. Line 242, information could be transffered to discussion). Our discussion about methods to prove apoptosis does not have to be reflected in the manuscript and there is no reason to be offended by questions.

 >> (Response) We would like express our sincere thanks for the reviewer's valuable time and diligent efforts dedicated to the thorough evaluation of our paper. We modifed page 7 (line 240-241) and page 12 (line 369-372)

Suggestions and comments:

Title – just to be precise and clear, it would be nice to point that the study is in PC cell LINES.

>> (Response) We changed the title according to your suggestion.

Abstract – please, see Line 34 – I strongly believe this study is about pancreatic, not prostate cancer!

 >> (Response) We modifed line 34.

Introduction – Line 85 – 87: author’s state that they “… illustrate the early morphological changes in the peritoneal metastasis”, however the main changes authors illustrate are molecular. In Line 86 authors state “… the connection between these morphological dynamics and their oncogenic properties was examined” – while I agree tha study shows molecular mechanism affected by GA, which experiments would subtantiate this statement? Which oncogenic properties were examined in this study?

 >> (Response) It must be a good topic for researching early epidemiology, but it will be another research area because there are no well-defined models for investigating the morphological and molecular features of pancreatic cancer cells. There have been recent publications on this topic [1, 2], and while I find it intriguing as a potential topic for future research. Thank you for your comments and sorry for the confusion. We modified the setences line 87-91.

Materials and Methods – perhaps it is a type-o in Line 93 that GA stock at 150 uM in EtOH was made, because it would be hard to use it to make 300 uM dilution for cell viability assay. Was it 150 mM?

 >> (Response) We modifed page 2 (line 93).

Another type-o is in Line 112 – double 300 uM .

 >> (Response) The double term 300 uM was removed.

Wording – please, choose one – cells planted or cells seeded (this would be better) and use it everywhere.

 >> (Response) We modifed page 3 (line 108 and 116).

Since GA was diluted in ethanol, did authors use a solvent control in the experiments?

 >> In accordance with the results showed in Figure 1, treatment with ethanol (0.25–1%) at 37 °C did not have any cytotoxicity[3]. We diluted it to 1/1000. In other words, it is not toxic at 0.1% because only about 1ul is added to 1000ml., which is an extremely small amount.

Results -  

Rev comment: a) why have not Annexin V staining performed for assessment of apoptosis level in cells?

Author’s response:  Thank you for your valuable comments. I understand your suggestion but we couldn’t perform the Annexin V because our FACs machine is out order during this study, so we couldn’t do so. However, we performed the TUNEL assay, cell cycle assay and apoptosis related protein analysis by Western blot analysis.

Rev response: I believe, the data for Figure 2.D.E. were obtained using FACS. Also, Annexin staining can be read by IF microscopy as well. However, I would like to close this part of our discussion. As pointed many times by you – yes, WB and differences in caspases and other apoptosis-related proteins are detected.

 >> (Response) Yes that is correct that FACs was used for cell cycle assay but the manchine is not available after the experiments. Thank you for understanding.

Discussion –

Rev comment: There are some confusing statements, e.g. Line 416 - 417: "The research study revealed the NO has inhibitory effect on apoptosis [meaning that more NO will lead to LESS apoptotic cells - rev.comment]. While these cells underwent apoptosis when exposed to high quantities of NO [66] [in this sentence authors contradict the previous sentence - rev.comment]."
 Author’s response:  We made modifications (page 13, line 417-418). 

  1. Jakštys, B., et al., Different Cell Viability Assays Reveal Inconsistent Results After Bleomycin Electrotransfer In Vitro. J Membr Biol, 2015. 248(5): p. 857-63.

Rev response: Please, read the paragraph – those 2 sentences (marked red in author’s revised manuscript) could be integrated better.  

  >> (Response) We modifed, page 13 line 415-418.

Comments on the Quality of English Language

The language is improved, but some last editing could help (see comments of type-o etc). 

  1. Odagiri, T., et al., The Cell Line-Dependent Diversity in Initial Morphological Dynamics of Pancreatic Cancer Cell Peritoneal Metastasis Visualized by an Artificial Human Peritoneal Model. Journal of Surgical Research, 2021. 261: p. 351-360.
  2. Asano, Y., et al., Construction of artificial human peritoneal tissue by cell-accumulation technique and its application for visualizing morphological dynamics of cancer peritoneal metastasis. Biochemical and biophysical research communications, 2017. 494(1-2): p. 213-219.
  3. Quintana, M., et al., Ethanol Enhances Hyperthermia-Induced Cell Death in Human Leukemia Cells. Int J Mol Sci, 2021. 22(9).
